# Independence Testing for Bounded Degree Bayesian Networks

**Arnab Bhattacharyya**
School of Computing
National University of Singapore
arnabb@nus.edu.sg

**Clément L. Canonne**
School of Computer Science
University of Sydney
clement.canonne@sydney.edu.au

**Joy Qiping Yang**
School of Computing
National University of Singapore
joy.yang@nus.edu.sg

## Abstract

We study the following *independence testing* problem: given access to samples from a distribution $P$ over $\{0, 1\}^n$, decide whether $P$ is a product distribution or whether it is $\varepsilon$-far in total variation distance from any product distribution. For arbitrary distributions, this problem requires $\exp(n)$ samples. We show in this work that if $P$ has a sparse structure, then in fact only linearly many samples are required. Specifically, if $P$ is Markov with respect to a Bayesian network whose underlying DAG has in-degree bounded by $d$, then $\tilde{\Theta}(2^{d/2} \cdot n/\varepsilon^2)$ samples are necessary and sufficient for independence testing.

## 1 Introduction

It is often convenient to model high-dimensional datasets as probability distributions. An important reason is that the language of probability formalizes what we intuitively mean by two features of the data being *independent*: the marginal distributions along the two corresponding coordinates are statistically independent. Independence is a basic probabilistic property that, when true, enables better interpretability of data as well as computationally fast inference algorithms.

In this work, we study the problem of testing whether a collection of $n$ binary random variables are mutually independent. Independence testing is an old and foundational problem in hypothesis testing [Neyman and Pearson, 1933, Lehmann et al., 2005], and we consider it from the perspective of *property testing* [Rubinfeld and Sudan, 1996, Goldreich et al., 1998]. Our goal is to design a tester that accepts product distributions on $n$ variables and rejects distributions that have statistical distance at least $\varepsilon$ from every product distribution. The tester gets access to i.i.d. samples from the input distribution and can fail (in either case) with probability at most $1/3$. The objective is to minimize the number of samples as a function of $n$ and $\varepsilon$.

Previous work [Diakonikolas and Kane, 2016] shows[1] that testing independence of a distribution on $\{0, 1\}^n$ requires $2^{\Omega(n)}$ queries which is intractable. However, there is an aspect of this proof that is unsatisfying. The "hard distributions", which are shown to require many queries to distinguish from product distributions, cannot be described succinctly, as they require exponentially many bits to describe. Thus, a natural question arises: can we test independence efficiently for distributions on $\{0, 1\}^n$ that have a sparse description?

---

[1] See Appendix D for a more general result.

36th Conference on Neural Information Processing Systems (NeurIPS 2022).

One of the most canonical ways to describe high-dimensional distributions is as *Bayesian networks* (or *Bayes nets* in short). A Bayes net specifies how to generate an $n$-dimensional sample in an iterative way and is especially useful for modeling causal relationships. Formally, a Bayes net on $\{0,1\}^n$ is given by a directed acyclic graph (DAG) $G$ on $n$ vertices and probability distributions $p_{i,\pi}$ on $\{0,1\}$ for all $i \in [n]$ and all assignments $\pi$ to the parents of the $i$'th node in the graph $G$. An $n$-dimensional sample is obtained by sampling the nodes in a topological order of $G$, where the $i$'th node is sampled according to $p_{i,\pi}$ for the assignment $\pi$ that is already fixed by the samples of the parent nodes of $i$. The generated distribution on $\{0,1\}^n$ is said to be *Markov with respect to $G$*.

In this work, we consider independence testing on distributions having a sparse Bayes net description, a class of distributions naturally arising in, and with numerous applications to, machine learning [Wainwright and Jordan, 2008], robotics, natural language processing, medicine, and biological settings such as gene expression data [Friedman et al., 2000, Peng et al., 2009, Gardner et al., 2003] (where it is known that most genetic networks are actually sparse). In these cases, one hopes to leverage that additional knowledge to test whether those sparse, local dependencies are actually present without having to pay the prohibitive exponential cost in the dimension $n$. Specifically, we analyze independence testing on distributions that are promised to be Markov with respect to a DAG of maximum in-degree $d$, where $d \ll n$. While the learning sample complexity, known to be $\tilde{O}(2^d \cdot n/\varepsilon^2)$ [Bhattacharyya et al., 2020], provides a baseline for the *testing* question, it is not at all obvious that this is tight, and what the correct dependence on $d$ and even $n$ are. Our main result essentially settles this question, and establishes the following:

**Theorem 1.1** (Informal Main Theorem). *Suppose an unknown distribution $P$ on $\{0,1\}^n$ is Markov with respect to an unspecified degree-$d$ DAG. The sample complexity of testing whether $P$ is a product distribution or is at least $\varepsilon$-far from any product distribution is $\tilde{\Theta}(2^{d/2} \cdot n/\varepsilon^2)$.*

In the course of proving this theorem, we additionally derive several technical results that are of independent interest, such as bounds on the moment generating function of squared binomials and an independence testing algorithm for arbitrary distributions in Hellinger distance.

Our work explicitly initiates the study of *testing graphical structure* in the context of property testing. That is, instead of testing a statistical property of a distribution (as is typical in distribution property testing), we can interpret our problem as that of testing a graphical property of the underlying graph that describes the distribution. In particular, testing independence can be viewed as testing maximum degree-0 of a distribution's graph. This point of view opens the door to testing many other relevant graphical properties of graphical models, e.g., maximum degree $k$, being a forest, being connected, etc. Hence, by analyzing the "base case" of independence testing, we provide a necessary first step to solving other graphical testing problems. Information-theoretic bounds for related problems were studied recently by Neykov et al. [2019].

## 1.1 Related Work

Distribution testing has been an active and rapidly progressing research program for the last 20+ years; see Rubinfeld [2012] and Canonne [2020] for surveys. One of the earliest works in this history was that of Batu et al. [2001] who studied testing independence of two random variables. There followed a series of papers [Alon et al., 2007, Levi et al., 2013, Acharya et al., 2015], strengthening and generalizing bounds for this problem, culminating in the work of Diakonikolas and Kane [2016] who gave tight bounds for testing independence of distributions over $[n_1] \times \cdots \times [n_d]$. Hao and Li [2020] recently considered the (harder) problem of estimating the distance to the closest product distribution (i.e., *tolerant* testing), showing this task could, too, be performed with a sublinear sample complexity.

Though most of the focus has been on testing properties of arbitrary input distributions, it has long been recognized that distributional restrictions are needed to obtain sample complexity improvements. For example, Rubinfeld and Servedio [2009], Adamaszek et al. [2010] studied testing uniformity of monotone distributions on the hypercube. Similarly, Daskalakis et al. [2012] considered the problem of testing monotonicity of $k$-modal distributions. The question of independence testing of structured high-dimensional distributions was considered in Daskalakis et al. [2019] in the context of Ising models. We note that while their work is in the same spirit as ours, Ising models and Bayes nets are incomparable modeling assumptions, and their results (and techniques) and ours are mostly disjoint. Further, while the results may overlap in some special cases, the conversion between

parameterizations makes them difficult to compare even in these cases (e.g., dependence on the maximum edge value parameter $\beta$ for Ising models, and max-undirected-degree vs. max-in-degree). More recently, Canonne et al. [2020], Daskalakis and Pan [2017], Bhattacharyya et al. [2020, 2021] have studied identity testing and closeness testing for distributions that are structured as degree-$d$ Bayes nets. Our work here continues this research direction in the context of independence testing.

## 1.2 Our techniques

*Upper bound.* The starting point of our upper bound is the (standard) observation that a distribution $P$ over $\{0,1\}^n$ is far from being a product if, and only if, it is far from the product of its marginals. By itself, this would not lead to any savings over the trivial exponential sample complexity. However, we can combine this with a localization result due to Daskalakis and Pan [2017], which then guarantees that if the degree-$d$ Bayes net $P$ is at *Hellinger* distance $\varepsilon$ from the product of its marginals $P'$, then there exists *some* vertex $i \in [n]$ such that $P_{i,\Pi_i}$ (the marginalization of $P$ onto the set of nodes consisting of $i$ and its $d$ parents) is at Hellinger distance at least $\frac{\varepsilon}{\sqrt{n}}$ from $P'_{i,\Pi_i}$. These two facts, combined, seem to provide exactly what is needed: indeed, given access to samples from $P$ and any fixed set of $d+1$ vertices $S$, one can simulate easily samples from both $P_S$ and $P'_S$ (for the second, using $d+1$ samples from $P$ to generate one from $P'_S$, as $P'_S$ is the product of marginals of $P_S$). A natural idea is then to iterate over all $\binom{n}{d+1}$ possible subsets $S$ of $d+1$ variables and check whether $P_S = P'_S$ for each of them using a closeness testing algorithm for arbitrary distributions over $\{0,1\}^{d+1}$: the overhead due to a union bound and the sampling process for $P_S, P'_S$ adds a factor $O(d \cdot \log \binom{n}{d+1}) = O(d^2 \log n)$ to the closeness testing procedure. However, since testing closeness over $\{0,1\}^{d+1}$ to total variation distance $\varepsilon'$ has sample complexity $O(2^{2d/3}/\varepsilon'^2)$ and, by the quadratic relation between Hellinger and total variation distances, we need to take $\varepsilon' = \frac{\varepsilon^2}{n}$, we would then expect the overall test to result in a $\tilde{O}(2^{2d/3}n^2/\varepsilon^4)$ sample complexity – much more than what we set out for.

A first natural idea to improve upon this is to use the refined identity testing result of Daskalakis and Pan [2017, Theorem 4.2] for Bayes nets, which avoids the back and forth between Hellinger and total variation distance and thus saves on this quadratic blowup. Doing so, we could in the last step keep $\varepsilon' = \frac{\varepsilon}{\sqrt{n}}$ (saving on this quadratic blowup), and pay only overall $\tilde{O}(2^{3d/4}/\varepsilon'^2) = \tilde{O}(2^{3d/4}n/\varepsilon^2)$. This is better, but still falls short of our original goal.

The second idea is to forego closeness testing in the last step entirely, and instead use directly an *independence* testing algorithm for arbitrary distributions over $\{0,1\}^{d+1}$, to test if $P_S$ is indeed a product distribution for every choice of $S$ considered. Unfortunately, while promising, this idea suffers from a similar drawback as our very first attempt: namely, the known independence testing algorithms are all designed for testing in total variation distance (not Hellinger)! Thus, even using an optimal TV testing algorithm for independence [Acharya et al., 2015, Diakonikolas and Kane, 2016] would still lead to this quadratic loss in the distance parameter $\varepsilon'$, and a resulting $\tilde{O}(2^{d/2}n^2/\varepsilon^4)$ sample complexity.

To combine the best of our last two approaches and achieve the claimed $\tilde{O}(2^{d/2}n/\varepsilon^2)$, we combine the two insights and perform, for each set $S$ of $d+1$ variables, independence testing on $P_S$ in Hellinger distance. In order to do so, however, we first need to design a testing algorithm for this task, as none was previously available in the literature. Fortunately for us, we are able to design such a testing algorithm (Lemma 3.2) achieving the desired – and optimal – sample complexity. Combining this Hellinger independence testing algorithm over $\{0,1\}^{d+1}$ with the above outline finally leads to the $\tilde{O}(2^{d/2}n/\varepsilon^2)$ upper bound of Theorem 1.1.

*Lower bound.* To obtain our $\Omega(2^{d/2}n/\varepsilon^2)$ lower bound on testing independence of a degree-$d$ Bayes net, we start with the construction introduced by Canonne et al. [2020] to prove an $\Omega(n/\varepsilon^2)$ sample complexity lower bound on testing *uniformity* of degree-1 Bayes nets. At a high level, this construction relies on picking uniformly at random a perfect matching $M$ of the $n$ vertices, which defines the structure of the Bayes net; and, for each of the $n/2$ resulting edges, picking either a positive or negative correlation (with value $\pm\varepsilon/\sqrt{n}$) between the two vertices, again uniformly at random. One can check relatively easily that every Bayes net $P_\lambda$ obtained this way, where $\lambda$ encodes the matching $M$ and the $2^{n/2}$ signs, is (1) a degree-1 Bayes net, (2) at total variation $\varepsilon$ from the uniform distribution $U$. The bulk of their analysis then lies in showing that (3) $\Omega(n/\varepsilon^2)$ samples are necessary to distinguish between $U$ and such a randomly chosen $P_\lambda$. Generalizing this lower bound

construction and analysis to independence testing (not just uniformity), and to degree-$d$ (and not just degree-1) Bayes nets turns out to be highly non-trivial, and is our main technical contribution.

Indeed, in view of the simpler and different $\Omega(2^{d/2}\sqrt{n}/\varepsilon^2)$ sample complexity lower bound for uniformity testing degree-$d$ Bayes nets obtained in Canonne et al. [2020] *when the structure of the Bayes net is known*,[2] one is tempted to adapt the same idea to the matching construction: that is, reserve $d-1$ out of the $n$ vertices to "encode" a pointer towards one of $2^{d-1}$ independently chosen $P_\lambda$'s as above (i.e., the hard instances are now uniform mixtures over $2^{d-1}$ independently generated degree-1 hard instances). The degree-1 hard instances lead in previous work to a tighter dependence on $n$ (linear instead of $\sqrt{n}$) because their Bayesian structure is *unknown*. Thus, by looking at the mixture of these degree-1 Bayes nets, one could hope to extend the analysis of that second lower bound from Canonne et al. [2020] and get the desired $\Omega(2^{d/2}n/\varepsilon^2)$ lower bound. Unfortunately, there is a major issue with this idea: namely, if the $2^{d-1}$ matchings are chosen independently, then the resulting overall distribution is unlikely to be a degree-$d$ Bayes net – instead, each vertex will have expected degree $\Omega(2^d)$! (This was not an issue in the corresponding lower bound of Canonne et al. [2020], since for them each of the $2^d$ components of the mixture was a degree-0 Bayes net, i.e., a product distribution; so degrees could not "add up" across the components.)

To circumvent this, we instead choose the matching $M$ to be common to all $2^{d-1}$ components of the mixture and only pick the sign of their $n/2$ correlations independently; thus ensuring that every node in the resulting distribution $P$ has degree $d$. This comes at a price, however: the analysis of the $\Omega(2^{d/2}\sqrt{n}/\varepsilon^2)$ lower bound from Canonne et al. [2020] crucially relied on independence across those components, and thus can no longer be extended to our case (where the $2^{d-1}$ distributions $P_\lambda$ share the same matching $M$, and thus we only have independence across components conditioned on $M$). Handling this requires entirely new ideas, and constitutes the core of our lower bound. In particular, from a technical point of view this requires us to handle the moment-generating-function of squares of Binomials, as well as that of (squared) truncated Binomials. To do so, we develop a range of results on Binomials and Multinomial distributions we believe are of independent interest.

Finally, after establishing that $\Omega(2^{d/2}n/\varepsilon^2)$ samples are necessary to distinguish the resulting "mixture of trees" $P$ from the uniform distribution $U$ (Lemma 2.3), it remains to show that this implies our stronger statement on testing *independence* (not just uniformity). To do so, we need to show that $P$ is not only far from $U$, but from *every* product distribution: doing so is itself far from immediate, and is established in Lemma 2.4 by relating the distance from the mixture $P$ to every product distribution to the distance between distinct components of the mixture, and lower bounding those directly by analyzing the concentration properties of each component $P_\lambda$ of our construction.

**Preliminaries.** We use the standard asymptotic notation $O(\cdot)$, $\Omega(\cdot)$ $\Theta(\cdot)$, and write $\tilde{O}(\cdot)$ to omit polylogarithmic factors in the argument. Throughout, we identify probability distributions over discrete sets with their probability mass functions (pmf), and further denote by $U$ (resp., $U_d$) the uniform distribution on $\{0,1\}^n$ (resp., $\{0,1\}^d$). We also write $P^{\otimes m}$ for the $m$-fold product of a distribution $P$, that is, $P \otimes \cdots \otimes P$ (the distribution of a tuple of $m$ i.i.d. samples from $P$); and $[n]$ for the set $\{1,\ldots,n\}$.

*Bayesian networks.* Given a directed acyclic graph (DAG) over $n$ nodes, a probability distribution $P$ over $\{0,1\}^n$ is said to be *Markov with respect to $G$* if $P$ factorizes according to $G$; we will also say that $P$ has *structure $G$*. A DAG $G$ is said to have in-degree $d$, if every node has at most $d$ parents; for convenience, we use *degree-$d$* exclusively as "in-degree $d$" throughout the paper; we will denote by $\Pi_i \subseteq [n]$ the set of parents of a node $i$. Finally, a distribution $P$ over $\{0,1\}^n$ is a *degree-$d$ Bayes net* if $P$ is Markov w.r.t. some degree-$d$ DAG.

*Distances between distributions.* Given two distributions $P, Q$ over the same (discrete) domain $\mathcal{X}$, the *total variation distance* (TV) between $P$ and $Q$ is defined as

$$d_{\mathrm{TV}}(P,Q) = \sup_{S \subseteq \mathcal{X}} \left( P(S) - Q(S) \right) = \frac{1}{2} \sum_{x \in \mathcal{X}}^{n} |P(x) - Q(x)| \in [0,1]. \tag{1}$$

While TV distance will be our main focus, we will also rely in our proofs on two other notions of distance between distributions: the *Hellinger distance*, given by $d_{\mathrm{H}}(P,Q) = \frac{1}{\sqrt{2}}\|\sqrt{P} - \sqrt{Q}\|_2$, and

---

[2] We note that generalizing this (weaker, in view of the dependence on $n$) $\Omega(2^{d/2}\sqrt{n}/\varepsilon^2)$ sample complexity lower bound to our setting is relatively simple, and we do so in Appendix D.1 (specifically, Theorem D.3).

the *chi-squared divergence*, defined by $d_{\chi^2}(P, Q) = \sum_{x \in \mathcal{X}} (P(x) - Q(x))^2/Q(x)$. TV distance, squared Hellinger distance, and $\chi^2$ divergence are all instances of $f$-divergences, and as such satisfy the data processing inequality; further, they are related by the following sequence of inequalities:

$$d_H^2(P, Q) \leqslant d_{TV}(P, Q) \leqslant \sqrt{2}d_H(P, Q) \leqslant \sqrt{d_{\chi^2}(P, Q)} \qquad (2)$$

*Tools from previous work.* We finally state results from the literature which we will rely upon.

**Corollary 1.2** (Daskalakis and Pan [2017, Corollary 2.4]). *Suppose $P$ and $Q$ are distributions on $\Sigma^n$ with common factorization structure*

$$P(x) = P_{X_1}(x_1) \prod_{i=2}^n P_{X_i|X_{\Pi_i}}(x_i|x_{\Pi_i}), \qquad Q(x) = Q_{X_1}(x_1) \prod_{i=2}^n Q_{X_i|X_{\Pi_i}}(x_i|x_{\Pi_i}), \quad x \in \Sigma^n$$

*where we assume the nodes are topologically ordered, and $\Pi_i$ is the set of parents of $i$. Then*

$$d_H^2(P, Q) \leqslant d_H^2(P_{X_1}, Q_{X_1}) + d_H^2(P_{X_2, X_{\Pi_2}}, Q_{X_2, X_{\Pi_2}}) + \cdots + d_H^2(P_{X_n, X_{\Pi_n}}, Q_{X_n, X_{\Pi_n}}).$$

*In particular, if $d_H^2(P, Q) \geqslant \varepsilon$ then there exists some $i$ such that $d_H^2(P_{X_i, X_{\Pi_i}}, Q_{X_i, X_{\Pi_i}}) \geqslant \frac{\varepsilon}{n}$.*

**Lemma 1.3** (Acharya et al. [2015, Lemma 4]). *There exists an efficient algorithm which, given samples from a distribution $P$ over $[n_1] \times \cdots \times [n_d]$, outputs a product distribution $Q$ over $[n_1] \times \cdots \times [n_d]$ such that, if $P$ is a product distribution, then $d_{\chi^2}(p, q) \leqslant \varepsilon^2$. with probability at least $5/6$. The algorithm uses $O((\sum_{\ell=1}^d n_\ell)/\varepsilon^2)$ samples from $P$.*

**Theorem 1.4** (Daskalakis et al. [2018, Theorem 1]). *There exists an efficient algorithm which, given samples from a distribution $P$ and a known reference distribution $Q$ over $[n]$, as well as a distance parameter $\varepsilon \in (0, 1]$, distinguishes between the cases (i) $d_{\chi^2}(P, Q) \leqslant \varepsilon^2$ and (ii) $d_H(P, Q) \geqslant \varepsilon$ with probability at least $5/6$. The algorithm uses $O(n^{1/2}/\varepsilon^2)$ samples from $P$.*

**Lemma 1.5** (Batu et al. [2001, Proposition 1]). *Let $P$ be a discrete distribution supported on $[n] \times [m]$, with marginals $P_1$ and $P_2$. Then, we have $\min_{Q_1, Q_2} d_{TV}(P, Q_1 \otimes Q_2) \geqslant \frac{1}{3}d_{TV}(P, P_1 \otimes P_2)$, where the minimum is taken over all distributions $Q_1$ over $[n]$ and $Q_2$ over $[m]$.*

## 2 The $\Omega(2^{d/2}n/\varepsilon^2)$ Lower Bound

In this section, we prove our main technical result: a lower bound on the sample complexity of testing whether a degree-$d$ Bayes net is a product distribution. In the interest of space, we omit some details of the derivations: the interested reader can find them in Appendix A.

**Theorem 2.1.** *Let $d \leqslant c \cdot \log n$, where $c > 0$ is a sufficiently small absolute constant. Then, testing whether an arbitrary degree-$d$ Bayes net over $\{0, 1\}^n$ is a product distribution or is $\varepsilon$-far from every product distribution requires $\Omega(2^{d/2}n/\varepsilon^2)$ samples.*

This theorem considerably generalizes the lower bound of $\Omega(n/\varepsilon^2)$ established in Canonne et al. [2020] for the case of trees[3] ($d = 1$). To establish our result, we build upon (and considerably extend) their analysis; in particular, we will rely on the following "mixture of trees" construction, which can be seen as a careful mixture of ($2^{d-1}$ of) the hard instances from the lower bound of [Canonne et al., 2020, Theorem 14]. Figure 1 shows an illustrative example of our lower bound constructions.

**Notation.** Throughout, for given $n, d, \varepsilon$, we let $N := n - d + 1$ (without loss of generality assumed to be even), $D := 2^{d-1}$, $\delta := \frac{\varepsilon}{\sqrt{n}}$, and

$$z_0 = \frac{1 + 4\delta}{1 - 4\delta}, \quad z_1 = \frac{1 + (4\delta)^2}{1 - (4\delta)^2}, \quad z_2 = \frac{1 + (4\delta)^4}{1 - (4\delta)^4}.$$

**Definition 2.2** (Mixture of Trees). Given parameters $0 \leqslant d \leqslant n$ and $\delta \in (0, 1]$, we define the probability distribution $\mathcal{D}_{n,d,\delta}$ over degree-$d$ Bayes nets by the following process.

---

[3]While this can be better visualized as a forest, we use the term "tree" to refer to the structure of degree-1 Bayes nets. Technically, it still factorizes as a Bayesian path (tree), but not all edges are necessary.

1. Choose a perfect matching $\lambda$ of $[N]$ uniformly at random (where $N = n - d + 1$), i.e., a set of $N/2$ disjoint pairs;
2. Draw i.i.d. $\mu_1, \ldots, \mu_D$ uniformly at random in $\{0, 1\}^{N/2}$;
3. For $\ell \in [D]$, let $p_\ell$ be the distribution over $\{0, 1\}^N$ defined as the Bayes net with tree structure $\lambda$, such that if $\lambda_k = (i, j) \in \lambda$ then the corresponding covariance between variables $X_i, X_j$ is
$$\mathrm{Cov}(X_i, X_j) = (-1)^{\mu_{\ell,k}} \delta$$
4. Let the resulting distribution $P_{\lambda,\mu}$ over $\{0, 1\}^n$ be

$$P_{\lambda,\mu}(x) = \frac{1}{D} \sum_{\ell=1}^{D} p_\ell(x_d, \ldots, x_n) \mathbb{1}[\iota(x_1, \ldots, x_{d-1}) = \ell]$$

where $\iota \colon \{0, 1\}^{d-1} \to [D]$ is the indexing function, mapping the binary representation to the corresponding number.

That is, $\mathcal{D}_{n,d,\delta}$ is the uniform distribution over the set of degree-$d$ Bayes nets where the first $d - 1$ coordinates form a "pointer" to one of the $2^{d-1}$ tree Bayes nets sharing the same tree structure (the matching $\lambda$), but with independently chosen covariance parameters (the $D$ parameters $\mu_1, \ldots, \mu_D$).

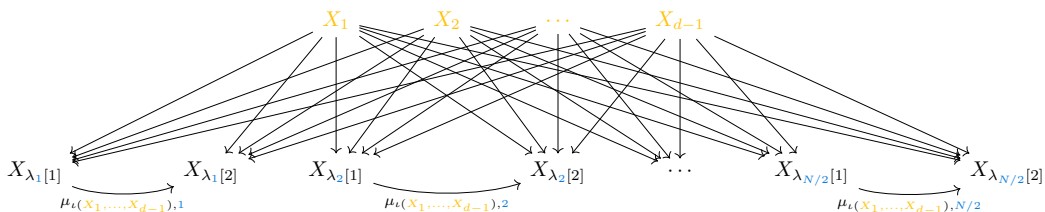

Figure 1: A depiction of our lower bound construction, where the arrows are the edges of the underlying Bayes net. The first $(d - 1)$ nodes can be seen as a $(d - 1)$-bit string that acts as a pointer to one of the distributions, i.e., the value of the first $d$ nodes encodes which distribution the rest of the nodes are on. We write $\lambda_k[1] := i$ and $\lambda_k[2] := j$ when $\lambda_k = (i, j)$. There are two sources of randomness in the construction. The first one is the random matching on the $N/2$ pairs represented in graph in the form of $(\lambda_1[1], \lambda_1[2]), \ldots, (\lambda_{N/2}[1], \lambda_{N/2}[2])$, which is independent of the particular configuration $X_1, \ldots, X_d$ takes. The second one is the covariance parameters between each pair; these parameters are randomly and independently chosen for each configuration (in total $2^d$ of them) of the first $d$ nodes. Color (yellow and blue) marks how the value of covariance parameter $\mu_{\ell,k}$ in each pair depends on the first $d$ nodes' configurations as well as the matching parameter $k$.

With this construction in hand, Theorem 2.1 will follow from the next two lemmas:

**Lemma 2.3** (Indistinguishability). *There exist absolute constants $c, C > 0$ such that the following holds. For $\Omega(1) \leqslant d \leqslant c \log n$, no $m$-sample algorithm can distinguish with probability at least $2/3$ between a (randomly chosen) mixture of trees $P \sim \mathcal{D}_{n,d,\varepsilon/\sqrt{n}}$ and the uniform distribution $U$ over $\{0, 1\}^n$, unless $m \geqslant C \cdot 2^{d/2} n / \varepsilon^2$.*

**Lemma 2.4** (Distance from product). *Fix any $0 \leqslant d \leqslant \frac{n}{2}$ and $\varepsilon \in (0, 1]$. With probability at least $9/10$ over the choice of $P \sim \mathcal{D}_{n,d,\varepsilon/\sqrt{n}}$, $P$ is $\Omega(\varepsilon)$-far from every product distribution on $\{0, 1\}^n$.*

Note that this guarantees farness from *every* product distribution, not just from the uniform distribution. We focus on the proof sketch of Lemma 2.3 in the next subsection, and defer the proof of Lemma 2.4 to the appendix A.2. Throughout, we fix $n$, $d$, and $\varepsilon \in (0, 1]$.

## 2.1 Sample Complexity to distinguish from Uniform (Lemma 2.3)

**Proof sketch.** Following the original analogous analysis, we first use Ingster's method to upper bound square TV distance in (4) and after a series of algebraic calculations, we arrive at (6). This is where we improve upon the original analysis by substituting a tighter upper bound (7). This leads us to some of the most technical portions of the paper. To upper bound the inner expectation of (8), we use Lemma 2.5 to bound the unwieldy expression (average over $z_j^{\kappa_{j,i}}$ and raised by $m$) by an expectation of a multinomial variable $\vec{\alpha}$ in the expression $\prod_{i=1}^{D} \prod_{j=1}^{K} \cosh(2\alpha_i \delta_j)$ in (9); then, with some careful analysis and some tools on MGF (Moment Generating Function) of Binomial and

Multinomial from Section B, we can upper bound the latter expression in (12), which finally gives us the sample complexity lower bound.

**Details.** We now proceed with the proof of Lemma 2.3, starting with some convenient notations; some of the technical lemmas and facts used here are stated and proven in Appendix B. Let $\theta = (\lambda, \mu_1, \ldots, \mu_{2^{d-1}})$, where each $\mu_k \in \{\pm 1\}^{N/2}$, and let $P_\theta$ be the distribution for the mixture of trees construction from Definition 2.2. The following denotes the *matching count* between $(\lambda, \mu)$ and $x$ as the quantity

$$c(\lambda, \mu, x) := \left| \left\{ (i, j) \in \{d, \ldots, n\}^2 : \exists 1 \leqslant k \leqslant \tfrac{N}{2}, \lambda_k = (i, j) \text{ and } (-1)^{x_i + x_j} = (-1)^{\mu_{\iota(x),k}} \right\} \right|.$$

We will also introduce an analogous quantity with an "offset", for $x_{\text{ch}} = (x_d, \ldots, x_n)$, referring exclusively to the child nodes of $x$ (i.e., the last $N$ nodes, which are the "children" of the first $d-1$ "pointer nodes" in our construction),

$$c_{\text{ch}}(\lambda, \mu, x_{\text{ch}}) := \tag{3}$$
$$\left| \left\{ (i, j) \in \{d, \ldots, n\}^2 : \exists 1 \leqslant k \leqslant \tfrac{N}{2}, \lambda_k = (i - d + 1, j - d + 1) \text{ and } (-1)^{x_i + x_j} = (-1)^{\mu_{\iota(x),k}} \right\} \right|.$$

To denote the parameters of the "mixture of trees", we write $\theta_i := (\lambda, \mu_i)$ (for $i \in [D]$), recalling that the matching parameter $\lambda$ is common to all $D$ tree components. Since each $\mu_i$ corresponds to one of the values of $(x_1, \ldots, x_{d-1}) \in \{0, 1\}^{d-1}$, we as before use $\iota \colon \{0, 1\}^{d-1} \to [D]$ to denote the indexing function (so that, for instance, $\iota(x_1 = \cdots = x_{d-1} = 0) = 0$). We finally introduce three more quantities, related to the matching and orientations parameters across the $D$ components of the mixture:

$$A_{\theta_i, \theta_i'} := \{(s, t) \in \{1, \ldots, N/2\}^2 : \lambda[s] = \lambda'[t], \mu_i[s] = \mu_i'[t]\}$$
$$\text{(common pairs, same orientation)}$$

$$B_{\theta_i, \theta_i'} := \{(s, t) \in \{1, \ldots, N/2\}^2 : \lambda[s] = \lambda'[t], \mu_i[s] \neq \mu_i'[t]\}$$
$$\text{(common pairs, different orientation)}$$

$$C_{\theta_i, \theta_i'} := (\lambda \cup \lambda') \setminus (A \cup B) \qquad \text{(pairs unique to } \theta \text{ or } \theta')$$

For ease of notation, we define $A_i := A_{\theta_i, \theta_i'}$, $B_i := B_{\theta_i, \theta_i'}$ and $C_i := C_{\theta_i, \theta_i'}$; and note that $C_i = C_1$, as it only depends on $\lambda$ (not on the orientation $\mu_i$).

To prove the indistinguishability, we will bound the squared total variation distance (or equivalently, squared $\ell_1$) distance between the distributions of $m$ samples from (the uniform mixture of) $P_\theta$ and $U$ by a small constant; that is, between $Q := \mathbb{E}_\theta[P_\theta^{\otimes m}]$ and $U^{\otimes m}$. From Ingster's method (see, e.g., [Acharya et al., 2020, Lemma III.8.]), by using chi-square divergence as an intermediate step we get

$$\left\| Q - U^{\otimes m} \right\|_1^2 \leqslant d_{\chi^2}(Q, U^{\otimes m}) = \mathbb{E}_{\theta, \theta'}[(1 + \tau(\theta, \theta'))^m] - 1, \tag{4}$$

where $\tau(\theta, \theta') := \mathbb{E}_{x \sim U}\left[\left(\frac{P_\theta(x) - U(x)}{U(x)}\right)\left(\frac{P_{\theta'}(x) - U(x)}{U(x)}\right)\right]$. In order to get a handle on this quantity $\tau(\theta, \theta')$, we start by writing the expression for the density $P_\theta$. For any $x \in \{0, 1\}^n$, recalling Item 4 of Definition 2.2, $P_\theta(x) = \frac{1}{2^n}(1 + 4\delta)^{c(\lambda, \mu_{\iota(x)}, x)}(1 - 4\delta)^{\frac{N}{2} - c(\lambda, \mu_{\iota(x)}, x)}$.

Substituting this in the definition of $\tau$, we get an expression depending on $z_0 := \frac{1+4\delta}{1-4\delta}$. and the probability-generating function (PGF) of $c_{\text{ch}}(\lambda, \mu, x)$. Observing that, as $x \sim U_N$ and for fixed $\lambda, \mu$ we have that $c_{\text{ch}}(\lambda, \mu, x) \sim \text{Bin}\left(\frac{N}{2}, \frac{1}{2}\right)$, we can then massage this expression to get

$$1 + \tau(\theta, \theta') = \frac{1}{D}\left\{ \sum_{i=1}^{D}(1 - 4\delta)^N \cdot z_0^{|B_i|}\left(\frac{1 + z_0^2}{2}\right)^{|A_i|} \prod_{\sigma_i : |\sigma_i| \geqslant 4} \mathbb{E}_{\alpha \sim \mathcal{B}(\sigma_i)}[z_0^\alpha] \right\}, \tag{5}$$

where the product is taken over all cycles in the multigraph $G_{\theta, \theta'}$ induced by the two matchings; and, given a cycle $\sigma$, $\mathcal{B}(\sigma)$ is the probability distribution defined as follows. Say that a cycle $\sigma$ is *even* (resp., *odd*) if the number of edges with weight 1 along $\sigma$ is even (resp., odd); that is, a cycle is even or odd depending on whether the number of negatively correlated pairs along the cycle is an even or odd number. If $\sigma$ is an *even cycle*, then $\mathcal{B}(\sigma)$ is a Binomial with parameters $|\sigma|$ and $1/2$, conditioned on taking even values. Similarly, if $\sigma$ is an *odd cycle*, $\mathcal{B}(\sigma)$ is a Binomial with parameters $|\sigma|$ and $1/2$, conditioned on taking odd values. It follows that $\mathbb{E}_{\alpha \sim \mathcal{B}(\sigma)}[z_0^\alpha]$ is given by the following expression.

$$\mathbb{E}_{\alpha \sim \mathcal{B}(\sigma)}[z_0^\alpha] = \begin{cases} \mathbb{E}_{\alpha \sim \text{Bin}\left(|\sigma|, \frac{1}{2}\right)}[z^\alpha \mid \alpha \text{ even}] = \frac{(1 + z_0)^{|\sigma|} + (1 - z_0)^{|\sigma|}}{2^{|\sigma|}} & \text{, if } \sigma \text{ is even} \\ \mathbb{E}_{\alpha \sim \text{Bin}\left(|\sigma|, \frac{1}{2}\right)}[z^\alpha \mid \alpha \text{ odd}] = \frac{(1 + z_0)^{|\sigma|} - (1 - z_0)^{|\sigma|}}{2^{|\sigma|}} & \text{, if } \sigma \text{ is odd} \end{cases}.$$

Denote $\mathcal{S}_{e,i} \coloneqq \mathcal{S}_e(\theta_i, \theta_i') = \{\sigma \in \mathrm{cycle}(\theta_i, \theta_i') : |\sigma| \geqslant 4, \sigma \text{ is even}\}$, and similarly $\mathcal{S}_{o,i}$ for odd cycles. We would drop $\lambda, \mu$ or $i$, when clear from context. We first expand $\mathbb{E}_{\alpha \sim \mathcal{B}(\sigma)}[z_0^\alpha]$ as follows:

$$\prod_{\sigma : |\sigma| \geqslant 4} \mathbb{E}_{\alpha \sim \mathcal{B}(\sigma)}[z_0^\alpha] = \frac{(1 + z_0)^{|C|}}{2^{|C|}} \prod_{\sigma \in \mathcal{S}_e} (1 + (-4\delta)^{|\sigma|}) \prod_{\sigma \in \mathcal{S}_o} (1 - (-4\delta)^{|\sigma|}). \qquad (6)$$

We now improve upon the analogous analysis from [Canonne et al., 2020, Claim 12] to obtain a better upper bound for the remaining terms; indeed, the bound they derived is $e^{O(\varepsilon^4/n)}$, which was enough for their purposes but not ours (since it does not feature any dependence on $d$). Crucially, we instead derive the below inequality whose proof can be found in Appendix C.1:

$$\prod_{\sigma \in \mathcal{S}_e} (1 + (-4\delta)^{|\sigma|}) \prod_{\sigma \in \mathcal{S}_o} (1 - (-4\delta)^{|\sigma|}) \leqslant e^{c' \frac{\varepsilon^5}{N^{3/2}}} \prod_{\sigma \in \mathcal{S}_e : |\sigma| = 4} (1 + (-4\delta)^{|\sigma|}) \prod_{\sigma \in \mathcal{S}_o : |\sigma| = 4} (1 - (-4\delta)^{|\sigma|}), \quad (7)$$

for some absolute constant $c' > 0$. We will have $e^{c' \cdot m \varepsilon^5 / N^{3/2}} \leqslant e^{128 m \varepsilon^2 / (\sqrt{D} n)}$ if $D = O(n/\varepsilon^6)$, and this restriction on $D$ is satisfied for the regime of parameters considered in our lower bound, $d = O(\log n)$. Letting $R \coloneqq |A_1| + |B_1| \leqslant N/2$ and $z_1 \coloneqq \frac{1 + (4\delta)^2}{1 - (4\delta)^2}$, with (6) and some calculations,

$$\mathbb{E}_{\theta, \theta'}[(1 + \tau(\theta, \theta'))^m]$$
$$= \mathbb{E}_{\lambda, \lambda'}\left[(1 - (4\delta)^2)^{mR} \mathbb{E}_{\vec{\mu}, \vec{\mu}'}\left[\left(\frac{1}{D} \sum_{i=1}^D z_1^{|A_i|} \prod_{\sigma \in \mathcal{S}_{e,i}} (1 + (-4\delta)^{|\sigma|}) \prod_{\sigma \in \mathcal{S}_{o,i}} (1 - (-4\delta)^{|\sigma|})\right)^m\right]\right]. \qquad (8)$$

Fix a pair $\lambda, \lambda'$; we have that the $|A_i|$'s are i.i.d. $\mathrm{Bin}(R, 1/2)$ random variables. We now introduce $R' \coloneqq |\{\sigma_1 : |\sigma_1| = 4\}| = |\{\sigma_i : |\sigma_i| = 4\}|$, which is the random variable denoting how many cycles have length exactly 4. In particular, we have $R' \leqslant \frac{N}{4}$, since $\sum_{\sigma : |\sigma| \geqslant 4} \sigma = |C| \leqslant N$; more specifically, we have $R' \leqslant \frac{N - 2R}{4} \leqslant \frac{N}{4}$. Further, define $\kappa_i$ as the number of cycles of length 4 which have an even total number of negative correlations; that is, the number of cycles $\sigma$ such that $\mu_i, \mu_i'$ impose either 0, 2, or 4 negatively correlated pairs along that cycle.

Since $\mu, \mu'$ are uniformly distributed, being odd or even each has probability $1/2$, and thus $\kappa_i \sim \mathrm{Bin}(R', \frac{1}{2})$. Moreover, while $\kappa_i$ and $A_i$ both depend on $\mu_i, \mu_i'$, they by definition depend on disjoint subsets of those two random variables: thus, because each correlation parameter is chosen independently, we have that $\kappa_i$ and $A_i$ are independent conditioned on $(R, R')$. Now, recalling our setting of $z_2 = \frac{1 + (4\delta)^4}{1 - (4\delta)^4}$ and fixing a realization of $R, R'$, we have

$$\mathbb{E}_{\vec{\mu}, \vec{\mu}'}\left[\left(\frac{1}{D} \sum_{i=1}^D z_1^{|A_i|} \prod_{\sigma \in \mathcal{S}_e(i) : |\sigma| = 4} (1 + (4\delta)^4) \prod_{\sigma_i \in \mathcal{S}_o(i) : |\sigma| = 4} (1 - (4\delta)^4)\right)^m\right]$$
$$= (1 - (4\delta)^4)^{mR'} \mathbb{E}_{\vec{\mu}, \vec{\mu}'}\left[\left(\frac{1}{D} \sum_{i=1}^D z_1^{|A_i|} z_2^{\kappa_i}\right)^m\right]$$
$$\leqslant (1 - (4\delta)^4)^{mR'} z_1^{\frac{mR}{2}} z_2^{\frac{mR'}{2}} \mathbb{E}_{\vec{\alpha}}\left[\prod_{i=1}^D (\cosh(2\alpha_i \delta^2))^R (\cosh(2\alpha_i \delta^4))^{R'}\right], \qquad (9)$$

where (9) follows from the following lemma:

**Lemma 2.5.** *There exists an absolute constant $\delta_0 \approx 0.96$ such that the following holds. Let $K \geqslant 1$ and $R_1, \ldots, R_K$ be integers, and $\delta_1, \ldots, \delta_K \in (0, \delta_0]$. Suppose that $\kappa_{j,1}, \ldots, \kappa_{j,D} \sim \mathrm{Bin}(R_j, \frac{1}{2})$, are i.i.d., and mutually independent across $1 \leqslant j \leqslant K$, and $z_j \coloneqq \frac{1 + \delta_j}{1 - \delta_j}$. Then*

$$\mathbb{E}\left[\left(\frac{1}{D} \sum_{i=1}^D \prod_{j=1}^K z_j^{\kappa_{j,i}}\right)^m\right] \leqslant \left(\prod_{j=1}^K z_j^{\frac{m}{2} R_j}\right) \mathbb{E}_{\vec{\alpha}}\left[\prod_{i=1}^D \prod_{j=1}^K \cosh(2\alpha_i \delta_j)\right],$$

*where $(\alpha_1, \ldots, \alpha_D)$ follows a multinomial distribution with parameters $m$ and $(1/D, \ldots, 1/D)$.*

We now focus on the expectation on the right (last factor of the RHS of (9)): using that $\cosh u \leqslant \min(e^{u^2/2}, e^u)$ for $u \geqslant 0$, it can be shown, setting $\Delta := 1/\delta^2 = n/\varepsilon^2$, that

$$\mathbb{E}_{\vec{\alpha}}\left[\prod_{i=1}^{D}(\cosh(2\alpha_i\delta^2))^R(\cosh(2\alpha_i\delta^4))^{R'}\right]$$

$$\leqslant \mathbb{E}\left[\prod_{i=1}^{D}e^{8\alpha_i\delta^2 R\mathbb{1}[\alpha_i > \Delta]}\right]^{\frac{1}{4}}\mathbb{E}\left[\prod_{i=1}^{D}e^{8\alpha_i^2\delta^4 R\mathbb{1}[\alpha_i \leqslant \Delta]}\right]^{\frac{1}{4}}\mathbb{E}\left[\prod_{i=1}^{D}e^{4\alpha_i^2\delta^8 R'}\right]^{\frac{1}{2}} \tag{10}$$

where the threshold $\Delta$ was chosen as the value for which the term realizing the minimum changes. We first bound the product of the last two expectations:

$$\mathbb{E}\left[\prod_{i=1}^{D}e^{8\alpha_i^2\delta^4 R\mathbb{1}[\alpha_i \leqslant \Delta]}\right]^{1/4}\mathbb{E}\left[\prod_{i=1}^{D}e^{4\alpha_i^2\delta^8 R'}\right]^{1/2} \leqslant \mathbb{E}\left[\prod_{i=1}^{D}e^{8\delta^4 R\min(\alpha_i^2,\Delta^2)}\right]^{1/4}\mathbb{E}\left[\prod_{i=1}^{D}e^{4\alpha_i^2\delta^8 R'}\right]^{1/2}$$

$$\leqslant (\mathbb{E}[e^{8\min(\alpha_j^2,\Delta^2)\delta^4 R}])^{D/4}(\mathbb{E}[e^{4\alpha_1^2\delta^8 R'}])^{D/2} \tag{11}$$

$$\leqslant \exp\left(32\delta^4\frac{m^2}{D}R\right)\exp\left(32\delta^8\frac{m^2}{D}R'\right). \tag{12}$$

where we applied negative association (see, e.g., Dubhashi and Ranjan [1996, Theorem 13]) on both expectations for (11); and then got (12) by Lemmas B.7 and B.3 (for the latter, noting that $tm = 2\delta^8 mR' \leqslant 1/16$; and, for the former, assuming with little loss of generality that $\varepsilon \leqslant 1/(4\sqrt{2})$). Applying Lemma B.6 to the first (remaining) factor of the LHS above as $8\delta^2 R \leqslant 4$ and $D \geqslant \Omega(1)$, we get

$$\mathbb{E}\left[\prod_{i=1}^{D}e^{8\alpha_i\delta^2 R\mathbb{1}[\alpha_i > \Delta]}\right]^{1/4}\mathbb{E}\left[\prod_{i=1}^{D}e^{8\alpha_i^2\delta^4 R\mathbb{1}[\alpha_i \leqslant \Delta]}\right]^{1/4}\mathbb{E}\left[\prod_{i=1}^{D}e^{4\alpha_i^2\delta^8 R'}\right]^{1/2} \leqslant (1+o(1))\exp(32C'^2 R),$$

recalling that $R' \leqslant N/4 \leqslant n/4$, and our assumption that $m \leqslant C'\sqrt{D}n/\varepsilon^2$. Combining (7), (8) and (9), what we showed is

$$\mathbb{E}_{\theta,\theta'}[(1+\tau(\theta,\theta'))^m] \leqslant (1+o(1))e^{128\frac{m\varepsilon^2}{\sqrt{D}n}}\mathbb{E}_{\lambda,\lambda'}\left[(1-(4\delta)^2)^{mR}(1-(4\delta)^4)^{mR'}z_1^{\frac{mR}{2}}z_2^{\frac{mR'}{2}}e^{32C'^2\cdot R}\right]$$

$$\leqslant (1+o(1))e^{128\cdot C'}\mathbb{E}_{\lambda,\lambda'}\left[e^{32C'^2\cdot R}\right],$$

where the equality follows from the definition of $z_1, z_2$. To conclude, we will use the fact that, for every $k \geqslant 0$, $\Pr[R > k] \leqslant \frac{1}{k!}$, which was established in Canonne et al. [2020, p.46]. By summation by parts, one can show that this implies $\mathbb{E}_R\left[e^{\alpha R}\right] \leqslant 1+(1-e^{-\alpha})(e^\alpha - 1) \xrightarrow{\alpha \to 0} 1+\alpha(e-1)$ for any $\alpha > 0$, and so, in our case, $\mathbb{E}_{\theta,\theta'}[(1+\tau(\theta,\theta'))^m] \leqslant (1+o(1))e^{128\cdot C'}(1+(1-e^{-32C'^2})(e^{e^{32C'^2}}-1))$ In particular, the RHS can be made arbitrarily close to 1 by choosing a small enough value for the constant $C'$ (in the bound for $m$). By (4), this implies the desired bound on $\|Q - U^{\otimes m}\|_1^2$, and thus establishes Lemma 2.3. $\square$

# 3 Upper Bound for Independence Testing

In this section, we establish the upper bound part of Theorem 1.1, stated below:

**Theorem 3.1.** *There is an algorithm (Algorithm 1) with the following guarantees: given $\varepsilon \in (0, 1]$ and sample access to an unknown degree-d Bayes net P, the algorithm takes $O(d^2 2^{d/2}n\log(n)/\varepsilon^2)$ samples from P, and distinguishes with probability at least $2/3$ between the cases (i) P is a product distribution, and (ii) P is at total variation distance at least $\varepsilon$ from every product distribution.*

---

**Algorithm 1:** Independence testing for degree-$d$ Bayes net

---

**Input :** Independent samples from a degree-$d$ Bayes net $P$ over $\{0,1\}^n$, $\varepsilon \in (0,1]$

Set $\delta \leftarrow \frac{1}{3\binom{n}{d+1}}$, $\varepsilon' \leftarrow \frac{\varepsilon}{\sqrt{2n}(1+\sqrt{d+1})}$, and $m \leftarrow O\left(\frac{2^{d/2}}{\varepsilon'^2}\log\frac{1}{\delta}\right)$

Take a multiset $S$ of $m$ samples from $P$, where $\delta \leftarrow 1/(3\binom{n}{d+1})$

**for** *every subset $T \subseteq [n]$ of $d+1$ nodes* **do**

> /* We can generate one sample from $P_T$ given one from $P$    */
> Use the $m$ samples from $S$ to generate $m$ i.i.d. samples from $P_T$
> Call Algorithm 2 with distance parameter $\varepsilon'$ and failure probability $\delta$, using the $m$ samples
>  from $P_T$ /* This is over $\{0,1\}^{d+1}$.                 */

**if** *all $\binom{n}{d+1}$ tests accepted* **then  return** *accept*
**return** *reject*

---

As discussed in the introduction, the key components of our algorithm are performing the testing in Hellinger distance, in order to use the subadditivity result of Daskalakis and Pan [2017]; and using as subroutine an independence testing algorithm under Hellinger distance. As no optimal tester for this latter task was known prior to our work, our first step is to derive such an algorithm by adapting the "testing-by-learning" framework of Acharya et al. [2015]. We emphasize that, in view of the relation $\frac{1}{2}d_{\mathrm{TV}}^2 \leqslant d_{\mathrm{H}}^2 \leqslant d_{\mathrm{TV}}$, this is a strictly stronger statement than the analogous known $O(2^{n/2}/\varepsilon^2)$-sample algorithm for testing independence under total variation distance [Diakonikolas and Kane, 2016, Acharya et al., 2015], which only implies an $O(2^{n/2}/\varepsilon^4)$ sample complexity for testing independence under Hellinger distance.

**Lemma 3.2.** *There exists an algorithm (Algorithm 2) with the following guarantees: given a parameter $\varepsilon \in (0,1]$ and sample access to an unknown distribution $P$ over $\{0,1\}^n$, it takes $O(2^{n/2}/\varepsilon^2)$ samples from $P$, and distinguishes with probability at least $2/3$ between the cases (i) $P$ is a product distribution, and (ii) $P$ is at Hellinger distance at least $\varepsilon$ from every product distribution.*

---

**Algorithm 2:** Hellinger independence testing for general distributions

---

**Input :** Independent samples from the target distribution over $\{0,1\}^n$, $\varepsilon \in (0,1]$

Set $m_1 \leftarrow O(n/\varepsilon^2)$, and $m_2 \leftarrow O(2^{n/2}/\varepsilon^2)$

Take two multisets $S_1$ and $S_2$ of samples with size $m_1$, $m_2$ respectively from $P$

$\hat{P} \leftarrow$ call the algorithm of Lemma 1.3 with the samples from $S_1$

Call the tester in Theorem 1.4 with samples from $S_2$, to test $P$ with reference to $\hat{P}$ with distance $\varepsilon$
**if** *tester accepted* **then  return** *accept*
**return** *reject*

---

We will next require the following lemma, whose proof is deferred to Appendix C.2.

**Lemma 3.3.** *Let $P$ be a distribution on $\{0,1\}^n$, and $P' = (\pi_1 P) \otimes \cdots \otimes (\pi_n P)$ be the product of marginals of $P$. Denoting by $\mathcal{Q}$ the set of all product distributions on $\{0,1\}^n$, we have $\min_{Q \in \mathcal{Q}} d_{\mathrm{H}}(P,Q) \geqslant \frac{1}{1+\sqrt{n}}d_{\mathrm{H}}(P,P')$.*

We now have all the elements to prove Theorem 3.1. As discussed in the introduction, the idea is the following: when $P$ is a product distribution, then the distribution induced by $P'$ on any subset of $d+1$ nodes is still a product distribution. However, by the above localization Corollary 1.2, when $P$ is far from every product distribution, we can localize the farness between $P$ and $Q$ in one of the subsets of $d+1$ nodes. We also know that $P'$, the product of marginals $P$, is not too bad of an approximation to the closest $Q$ in the product space as suggested by Lemma 3.3. Thus, testing independence using $P'$ as a proxy and paying an extra factor of $O(d)$ to compensate in accuracy suffice. Details can be found in Appendix C.1.

**Acknowledgments.** Yang would like to thank Vipul Arora and Philips George John for the helpful discussions on the lower bound analysis; and Vipul, specifically, for providing valuable feedback on the manuscript. Arnab and Yang were partially supported by an NRF Fellowship for AI (NRF-NRFAI11-2019-0002). This work was done in part while Arnab was visiting the Simons Institute for the Theory of Computing.

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
