# A    Detailed version of Section 2

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

$$
\begin{aligned}
P_\theta(x) &= P_\theta(x_d, \ldots, x_n \mid x_1, \ldots, x_{d-1}) U_{d-1}(x_1, \ldots, x_{d-1}) \\
&= \frac{1}{2^{d-1}} P_{\lambda, \mu_{\iota(x)}}(x_d, \ldots, x_n) \\
&= \frac{1}{2^{d-1}} \cdot \frac{1}{2^N} (1 + 4\delta)^{c(\lambda, \mu_{\iota(x)}, x)} (1 - 4\delta)^{\frac{N}{2} - c(\lambda, \mu_{\iota(x)}, x)} \\
&= \frac{1}{2^n} (1 + 4\delta)^{c(\lambda, \mu_{\iota(x)}, x)} (1 - 4\delta)^{\frac{N}{2} - c(\lambda, \mu_{\iota(x)}, x)}.
\end{aligned}
$$

Substituting this in the definition of $\tau$, we get

$$
\begin{aligned}
\tau(\theta, \theta') &= \mathbb{E}_{x \sim U}\left[ \left( \frac{P_\theta(x)}{U(x)} - 1 \right) \left( \frac{P_{\theta'}(x)}{U(x)} - 1 \right) \right] \\
&= \mathbb{E}_{x \sim U}\left[ \left( (1 - 4\delta)^{\frac{N}{2}} \left( \frac{1 + 4\delta}{1 - 4\delta} \right)^{c(\lambda, \mu_{\iota(x)}, x)} - 1 \right) \left( (1 - 4\delta)^{\frac{N}{2}} \left( \frac{1 + 4\delta}{1 - 4\delta} \right)^{c(\lambda', \mu_{\iota(x)}', x)} - 1 \right) \right] \\
&= 1 + (1 - 4\delta)^N \mathbb{E}_{x \sim U}\left[ z_0^{c(\lambda, \mu_{\iota(x)}, x) + c(\lambda', \mu_{\iota(x)}', x)} \right] \\
&\quad - (1 - 4\delta)^{\frac{N}{2}} \mathbb{E}_{x \sim U}\left[ z_0^{c(\lambda, \mu_{\iota(x)}, x)} \right] - (1 - 4\delta)^{\frac{N}{2}} \mathbb{E}_{x \sim U}\left[ z_0^{c(\lambda', \mu_{\iota(x)}', x)} \right],
\end{aligned} \tag{15}
$$

where $z_0 := \frac{1+4\delta}{1-4\delta}$. As $x \sim U_N$, for fixed $\lambda, \mu$ we have that $c_{\mathrm{ch}}(\lambda, \mu, x)$ follows a $\mathrm{Bin}\left(\frac{N}{2}, \frac{1}{2}\right)$ distribution; recalling the expression of the Binomial distribution's probability generating function, we then have

$$
\begin{aligned}
(1-4\delta)^{\frac{N}{2}} \mathbb{E}_{x \sim U}[z_0^{c(\lambda, \mu_{\iota(x)}, x)}] &= (1-4\delta)^{\frac{N}{2}} \mathbb{E}_{\vec{x}_1 \sim U_{d-1}} \left[ \mathbb{E}_{\vec{x}_2 \sim U_N} \left[ z_0^{c_{\mathrm{ch}}(\lambda, \mu_{\iota(\vec{x}_1)}, \vec{x}_2)} \right] \right] \\
&= (1-4\delta)^{\frac{N}{2}} \frac{1}{D} \left\{ \sum_{i=1}^{D} \mathbb{E}_{m \sim \mathrm{Bin}\left(\frac{N}{2}, \frac{1}{2}\right)}[z_0^m] \right\} \\
&= (1-4\delta)^{\frac{N}{2}} \left( \frac{1+z_0}{2} \right)^{\frac{N}{2}} = 1.
\end{aligned}
$$

Using this to simplify the last two terms of (15), we obtain

$$
\begin{aligned}
1 + \tau(\theta, \theta') &= (1-4\delta)^N \mathbb{E}_{x \sim U}[z_0^{c(\lambda, \mu_{\iota(x)}, x) + c(\lambda', \mu'_{\iota(x)}, x)}] \\
&= \frac{1}{D} \left\{ \sum_{i=1}^{D} (1-4\delta)^N \mathbb{E}_{x \sim U_N}[z_0^{c_{\mathrm{ch}}(\lambda, \mu_i, x) + c_{\mathrm{ch}}(\lambda', \mu'_i, x)}] \right\} \\
&= \frac{1}{D} \left\{ \sum_{i=1}^{D} (1-4\delta)^N \cdot z_0^{|B_i|} \mathbb{E}_{\alpha \sim \mathrm{Bin}\left(|A_i|, \frac{1}{2}\right)}[z_0^{2\alpha}] \prod_{\sigma_i : |\sigma_i| \geqslant 4} \mathbb{E}_{\alpha \sim \mathcal{B}(\sigma_i)}[z_0^\alpha] \right\} \\
&= \frac{1}{D} \left\{ \sum_{i=1}^{D} (1-4\delta)^N \cdot z_0^{|B_i|} \left( \frac{1+z_0^2}{2} \right)^{|A_i|} \prod_{\sigma_i : |\sigma_i| \geqslant 4} \mathbb{E}_{\alpha \sim \mathcal{B}(\sigma_i)}[z_0^\alpha] \right\}, \qquad (16)
\end{aligned}
$$

where the product is taken over all cycles in the multigraph $G_{\theta, \theta'}$ induced by the two matchings; and, given a cycle $\sigma$, $\mathcal{B}(\sigma)$ is the probability distribution defined as follows. Say that a cycle $\sigma$ is *even* (resp., *odd*) if the number of edges with weight 1 along $\sigma$ is even (resp., odd); that is, a cycle is even or odd depending on whether the number of negatively correlated pairs along the cycle is an even or odd number. If $\sigma$ is an *even cycle*, then $\mathcal{B}(\sigma)$ is a Binomial with parameters $|\sigma|$ and $1/2$, conditioned on taking even values. Similarly, if $\sigma$ is an *odd cycle*, $\mathcal{B}(\sigma)$ is a Binomial with parameters $|\sigma|$ and $1/2$, conditioned on taking odd values. It follows that $\mathbb{E}_{\alpha \sim \mathcal{B}(\sigma)}[z_0^\alpha]$ is given by the following expression.

$$
\mathbb{E}_{\alpha \sim \mathcal{B}(\sigma)}[z_0^\alpha] = \begin{cases} \mathbb{E}_{\alpha \sim \mathrm{Bin}\left(|\sigma|, \frac{1}{2}\right)}[z^\alpha \mid \alpha \text{ even}] = \frac{(1+z_0)^{|\sigma|} + (1-z_0)^{|\sigma|}}{2^{|\sigma|}} & , \text{if } \sigma \text{ is even} \\ \mathbb{E}_{\alpha \sim \mathrm{Bin}\left(|\sigma|, \frac{1}{2}\right)}[z^\alpha \mid \alpha \text{ odd}] = \frac{(1+z_0)^{|\sigma|} - (1-z_0)^{|\sigma|}}{2^{|\sigma|}} & , \text{if } \sigma \text{ is odd} \end{cases}.
$$

Denote

$$
\begin{aligned}
\mathcal{S}_{e,i} &:= \mathcal{S}_e(\theta_i, \theta'_i) = \{\sigma \in \mathrm{cycle}(\theta_i, \theta'_i) : |\sigma| \geqslant 4, \sigma \text{ is even}\}, \\
\mathcal{S}_{o,i} &:= \mathcal{S}_o(\theta_i, \theta'_i) = \{\sigma \in \mathrm{cycle}(\theta_i, \theta'_i) : |\sigma| \geqslant 4, \sigma \text{ is odd}\}.
\end{aligned}
$$

We will often drop $\lambda, \mu$ or $i$, when clear from context. We expand $\mathbb{E}_{\alpha \sim \mathcal{B}(\sigma)}[z_0^\alpha]$ as follows:

$$
\begin{aligned}
\prod_{\sigma : |\sigma| \geqslant 4} \mathbb{E}_{\alpha \sim \mathcal{B}(\sigma)}[z_0^\alpha] &= \prod_{\sigma : |\sigma| \geqslant 4, \text{even}} \frac{(1+z_0)^{|\sigma|} + (1-z_0)^{|\sigma|}}{2^{|\sigma|}} \prod_{\sigma : |\sigma| \geqslant 4, \text{odd}} \frac{(1+z_0)^{|\sigma|} - (1-z_0)^{|\sigma|}}{2^{|\sigma|}} \\
&= \prod_{\substack{\sigma : |\sigma| \geqslant 4 \\ \text{even}}} \frac{(1+z_0)^{|\sigma|}}{2^{|\sigma|}} \left( 1 + \left( \frac{1-z_0}{1+z_0} \right)^{|\sigma|} \right) \prod_{\substack{\sigma : |\sigma| \geqslant 4 \\ \text{odd}}} \frac{(1+z_0)^{|\sigma|}}{2^{|\sigma|}} \left( 1 - \left( \frac{1-z_0}{1+z_0} \right)^{|\sigma|} \right) \\
&= \prod_{\sigma : |\sigma| \geqslant 4} \frac{(1+z_0)^{|\sigma|}}{2^{|\sigma|}} \prod_{\sigma \in \mathcal{S}_e} (1 + (-4\delta)^{|\sigma|}) \prod_{\sigma \in \mathcal{S}_o} (1 - (-4\delta)^{|\sigma|}) \\
&= \frac{(1+z_0)^{\sum_{\sigma : |\sigma| \geqslant 4} |\sigma|}}{2^{\sum_{\sigma : |\sigma| \geqslant 4} |\sigma|}} \prod_{\sigma \in \mathcal{S}_e} (1 + (-4\delta)^{|\sigma|}) \prod_{\sigma \in \mathcal{S}_o} (1 - (-4\delta)^{|\sigma|}) \\
&= \frac{(1+z_0)^{|C|}}{2^{|C|}} \prod_{\sigma \in \mathcal{S}_e} (1 + (-4\delta)^{|\sigma|}) \prod_{\sigma \in \mathcal{S}_o} (1 - (-4\delta)^{|\sigma|})
\end{aligned}
$$

where for the last equality we used that $\sum_{\sigma:|\sigma|\geqslant 4}|\sigma|=|C|$. We now improve upon the analogous analysis from [Canonne et al., 2020, Claim 12] to obtain a better upper bound for the remaining terms; indeed, the bound they derived is $e^{O(\varepsilon^4/n)}$, which was enough for their purposes but not ours (since it does not feature any dependence on $d$). Let $z_1 := \frac{1+(4\delta)^2}{1-(4\delta)^2}$. In view of using the above expression to bound (16), we first simplify (part of) the summands of (16) by using the fact that $2|A_i|+2|B_i|+|C_i|=N$ for all $i$, and following the same computations as in Canonne et al. [2020]:

$$(1-4\delta)^N z_0^{|B_i|}\left(\frac{1+z_0^2}{2}\right)^{|A_i|}\frac{(1+z_0)^{|C_i|}}{2^{|C_i|}}$$

$$=(1-4\delta)^N z_0^{|B_i|}\left(\frac{1+z_0^2}{2}\right)^{|A_i|}\frac{(1+z_0)^{|C_i|}}{2^{|C_i|}}$$

$$=((1-4\delta)^2 z_0)^{|B_i|}\left((1-4\delta)^2\frac{1+z_0^2}{2}\right)^{|A_i|}\underbrace{\left((1-4\delta)\frac{1+z_0}{2}\right)^{|C_i|}}_{=1}$$

$$=(1-(4\delta)^2)^{|B_i|}(1+(4\delta)^2)^{|A_i|}$$

$$=(1-(4\delta)^2)^{|A_i|+|B_i|}z_1^{|A_i|}=(1-(4\delta)^2)^{|A_1|+|B_1|}z_1^{|A_i|},$$

where the last equality uses the fact that the sum $|A_i|+|B_i|$ only depends on the matchings $\lambda,\lambda'$ (not the orientations $\mu_i,\mu_i'$), and thus is independent of $i$. Plugging this simplification into (16), and letting $R := |A_1|+|B_1|\leqslant N/2$ for convenience, we get

$$1+\tau(\theta,\theta')=\frac{1}{D}\left\{\sum_{i=1}^{D}(1-4\delta)^N\cdot z_0^{|B_i|}\left(\frac{1+z_0^2}{2}\right)^{|A_i|}\prod_{\sigma_i}\mathbb{E}_{\alpha\sim\mathcal{B}(\sigma_i)}[z_0^\alpha]\right\}$$

$$=(1-(4\delta)^2)^R\cdot\frac{1}{D}\left\{\sum_{i=1}^{D}z_1^{|A_i|}\prod_{\sigma\in\mathcal{S}_{e,i}}(1+(-4\delta)^{|\sigma|})\prod_{\sigma\in\mathcal{S}_{o,i}}(1-(-4\delta)^{|\sigma|})\right\}.$$

Next, we compute the expectation after raising the above to the power $m$.

$$\mathbb{E}_{\theta,\theta'}[(1+\tau(\theta,\theta'))^m]$$

$$=\mathbb{E}_{\theta,\theta'}\left[\left((1-(4\delta)^2)^R\frac{1}{D}\left\{\sum_{i=1}^{D}z_1^{|A_i|}\prod_{\sigma\in\mathcal{S}_{e,i}}(1+(-4\delta)^{|\sigma|})\prod_{\sigma\in\mathcal{S}_{o,i}}(1-(-4\delta)^{|\sigma|})\right\}\right)^m\right]$$

$$=\mathbb{E}_{\lambda,\lambda'}\left[(1-(4\delta)^2)^{mR}\mathbb{E}_{\vec{\mu},\vec{\mu}'}\left[\left(\frac{1}{D}\left\{\sum_{i=1}^{D}z_1^{|A_i|}\prod_{\sigma\in\mathcal{S}_{e,i}}(1+(-4\delta)^{|\sigma|})\prod_{\sigma\in\mathcal{S}_{o,i}}(1-(-4\delta)^{|\sigma|})\right\}\right)^m\right]\right].$$

$$\tag{17}$$

The quantity inside the inner expectation is quite unwieldy; to proceed, we will rely on the following identity, which lets us bound the two product terms:

$$\prod_{\sigma\in\mathcal{S}_e}(1+(-4\delta)^{|\sigma|})\prod_{\sigma\in\mathcal{S}_o}(1-(-4\delta)^{|\sigma|})\leqslant e^{c'\frac{\varepsilon^5}{N^{3/2}}}\prod_{\sigma\in\mathcal{S}_e:|\sigma|=4}(1+(-4\delta)^{|\sigma|})\prod_{\sigma\in\mathcal{S}_o:|\sigma|=4}(1-(-4\delta)^{|\sigma|}),\tag{18}$$

for some absolute constant $c'>0$. We defer the proof of this inequality to Appendix C.4, and proceed assuming it. Note that as long as $D=O(n/\varepsilon^6)$, we will have $e^{c'\cdot m\varepsilon^5/N^{3/2}}\leqslant e^{128m\varepsilon^2/(\sqrt{D}n)}$[4] and this restriction on $D$ is satisfied for the regime of parameters considered in our lower bound, $d=O(\log n)$.

Fix a pair $\lambda,\lambda'$; we have that the $|A_i|$'s are i.i.d. $\mathrm{Bin}(R,1/2)$ random variables. We now introduce

$$R':=|\{\sigma_1:|\sigma_1|=4\}|=|\{\sigma_i:|\sigma_i|=4\}|,$$

---

[4]As per the condition set in Lemma B.5, we will from now on assume that $n/\varepsilon^2\geqslant n\geqslant 40D$, which gives us $N^{3/2}\geqslant(n/2)^{3/2}\geqslant\frac{n}{2}\cdot(20D)^{1/2}\geqslant 2n\sqrt{D}$; and some more calculations give us $c'\frac{m\varepsilon^2}{N^{3/2}}\leqslant\frac{128m\varepsilon^2}{\sqrt{D}n}$.

which is the random variable denoting how many cycles have length exactly 4. In particular, we have $R' \leqslant \frac{N}{4}$, since $\sum_{\sigma:|\sigma|\geqslant 4} \sigma = |C| \leqslant N$; more specifically, we have $R' \leqslant \frac{N-2R}{4} \leqslant \frac{N}{4}$. Further, define $\kappa_i$ as the number of cycles of length 4 which have an even total number of negative correlations; that is, the number of cycles $\sigma$ such that $\mu_i, \mu_i'$ impose either 0, 2, or 4 negatively correlated pairs along that cycle.

Since $\mu, \mu'$ are uniformly distributed, being odd or even each has probability $1/2$, and thus $\kappa_i \sim \mathrm{Bin}\big(R', \frac{1}{2}\big)$. Moreover, while $\kappa_i$ and $A_i$ both depend on $\mu_i, \mu_i'$, they by definition depend on disjoint subsets of those two random variables: thus, because each correlation parameter is chosen independently, we have that $\kappa_i$ and $A_i$ are independent conditioned on $(R, R')$. Now, recalling our setting of $z_2 = \frac{1+(4\delta)^4}{1-(4\delta)^4}$ and fixing a realization of $R, R'$, we have

$$\mathbb{E}_{\vec{\mu},\vec{\mu}'}\left[\left(\frac{1}{D}\sum_{i=1}^{D} z_1^{|A_i|} \prod_{\sigma \in \mathcal{S}_e(i):|\sigma|=4}(1+(4\delta)^4) \prod_{\sigma_i \in \mathcal{S}_o(i):|\sigma|=4}(1-(4\delta)^4)\right)^m\right]$$

$$= \mathbb{E}_{\vec{\mu},\vec{\mu}'}\left[\left(\frac{1}{D}\sum_{i=1}^{D} z_1^{|A_i|}(1+(4\delta)^4)^{\kappa_i}(1-(4\delta)^4)^{R'-\kappa_i}\right)^m\right]$$

$$= (1-(4\delta)^4)^{mR'}\mathbb{E}_{\vec{\mu},\vec{\mu}'}\left[\left(\frac{1}{D}\sum_{i=1}^{D} z_1^{|A_i|} z_2^{\kappa_i}\right)^m\right]$$

$$\leqslant (1-(4\delta)^4)^{mR'} z_1^{\frac{mR}{2}} z_2^{\frac{mR'}{2}}\mathbb{E}_{\vec{\alpha}}\left[\prod_{i=1}^{D}(\cosh(2\alpha_i\delta^2))^R(\cosh(2\alpha_i\delta^4))^{R'}\right], \qquad (19)$$

where (19) follows from the following lemma, whose proof we defer to the end of the section:

**Lemma 2.5.** *There exists an absolute constant $\delta_0 \approx 0.96$ such that the following holds. Let $K \geqslant 1$ and $R_1, \ldots, R_K$ be integers, and $\delta_1, \ldots, \delta_K \in (0, \delta_0]$. Suppose that $\kappa_{j,1}, \ldots, \kappa_{j,D} \sim \mathrm{Bin}\big(R_j, \frac{1}{2}\big)$, are i.i.d., and mutually independent across $1 \leqslant j \leqslant K$, and $z_j := \frac{1+\delta_j}{1-\delta_j}$. Then*

$$\mathbb{E}\left[\left(\frac{1}{D}\sum_{i=1}^{D}\prod_{j=1}^{K} z_j^{\kappa_{j,i}}\right)^m\right] \leqslant \left(\prod_{j=1}^{K} z_j^{\frac{m}{2}R_j}\right)\mathbb{E}_{\vec{\alpha}}\left[\prod_{i=1}^{D}\prod_{j=1}^{K}\cosh(2\alpha_i\delta_j)\right],$$

*where $(\alpha_1, \ldots, \alpha_D)$ follows a multinomial distribution with parameters $m$ and $(1/D, \ldots, 1/D)$.*

We now focus on the expectation on the right (last factor of the RHS of (19)): using that $\cosh u \leqslant \min(e^{u^2/2}, e^u)$ for $u \geqslant 0$, we have, setting $\Delta := 1/\delta^2 = n/\varepsilon^2$,

$$\mathbb{E}_{\vec{\alpha}}\left[\prod_{i=1}^{D}(\cosh(2\alpha_i\delta^2))^R(\cosh(2\alpha_i\delta^4))^{R'}\right]$$

$$\leqslant \mathbb{E}_{\vec{\alpha}}\left[\prod_{i=1}^{D}\min(e^{2\alpha_i\delta^2 R}, e^{2\alpha_i^2\delta^4 R})e^{2\alpha_i^2\delta^8 R'}\right]$$

$$\leqslant \mathbb{E}_{\vec{\alpha}}\left[\prod_{i=1}^{D} e^{2\alpha_i\delta^2 R\mathbb{1}[\alpha_i>\Delta]}e^{2\alpha_i^2\delta^4 R\mathbb{1}[\alpha_i\leqslant\Delta]}e^{2\alpha_i^2\delta^8 R'}\right]$$

$$\leqslant \mathbb{E}\left[\prod_{i=1}^{D} e^{8\alpha_i\delta^2 R\mathbb{1}[\alpha_i>\Delta]}\right]^{1/4}\mathbb{E}\left[\prod_{i=1}^{D} e^{8\alpha_i^2\delta^4 R\mathbb{1}[\alpha_i\leqslant\Delta]}\right]^{1/4}\mathbb{E}\left[\prod_{i=1}^{D} e^{4\alpha_i^2\delta^8 R'}\right]^{1/2} \qquad (20)$$

where the last step comes from the generalized Hölder inequality (or, equivalently, two applications of the Cauchy–Schwarz inequality), and the threshold $\Delta$ was chosen as the value for which the term realizing the minimum changes. We first bound the product of the last two expectations:

$$\mathbb{E}\left[\prod_{i=1}^{D} e^{8\alpha_i^2\delta^4 R\mathbb{1}[\alpha_i\leqslant\Delta]}\right]^{1/4}\mathbb{E}\left[\prod_{i=1}^{D} e^{4\alpha_i^2\delta^8 R'}\right]^{1/2}$$

$$\leqslant \mathbb{E}\left[\prod_{i=1}^{D} e^{8\delta^4 R \min(\alpha_i^2, \Delta^2)}\right]^{1/4} \mathbb{E}\left[\prod_{i=1}^{D} e^{4\alpha_i^2 \delta^8 R'}\right]^{1/2}$$

$$\leqslant (\mathbb{E}[e^{8\min(\alpha_j^2, \Delta^2)\delta^4 R}])^{D/4} (\mathbb{E}[e^{4\alpha_1^2 \delta^8 R'}])^{D/2} \tag{21}$$

$$\leqslant \exp\left(32\delta^4 \frac{m^2}{D} R\right) \exp\left(32\delta^8 \frac{m^2}{D} R'\right). \tag{22}$$

where we applied negative association (see, e.g., Dubhashi and Ranjan [1996, Theorem 13]) on both expectations for (21); and then got (22) by Lemmas B.7 and B.3 (for the latter, noting that $tm = 2\delta^8 m R' \leqslant 1/16$; and, for the former, assuming with little loss of generality that $\varepsilon \leqslant 1/(4\sqrt{2})$). Applying Lemma B.6 to the first (remaining) factor of the LHS above as $8\delta^2 R \leqslant 4$ and $D \geqslant \Omega(1)$, we get

$$\mathbb{E}\left[\prod_{i=1}^{D} e^{8\alpha_i \delta^2 R \mathbb{1}[\alpha_i > \Delta]}\right]^{1/4} \mathbb{E}\left[\prod_{i=1}^{D} e^{8\alpha_i^2 \delta^4 R \mathbb{1}[\alpha_i \leqslant \Delta]}\right]^{1/4} \mathbb{E}\left[\prod_{i=1}^{D} e^{4\alpha_i^2 \delta^8 R'}\right]^{1/2}$$

$$\leqslant (1 + o(1)) \cdot \exp\left(32\delta^4 \frac{m^2}{D} R\right) \exp\left(32 \frac{m^2}{D} \delta^8 R'\right)$$

$$= (1 + o(1)) \exp(32 C'^2 R),$$

recalling that $R' \leqslant N/4 \leqslant n/4$, and our assumption that $m \leqslant C'\sqrt{D}n/\varepsilon^2$. Combining (17), (18), and (19), what we showed is

$$\mathbb{E}_{\theta, \theta'}[(1 + \tau(\theta, \theta'))^m] \leqslant (1 + o(1)) e^{128 \frac{m\varepsilon^2}{\sqrt{D}n}} \mathbb{E}_{\lambda, \lambda'}\left[(1 - (4\delta)^2)^{mR} (1 - (4\delta)^4)^{mR'} z_1^{\frac{mR}{2}} z_2^{\frac{mR'}{2}} e^{32 C'^2 \cdot R}\right]$$

$$= (1 + o(1)) e^{128 \cdot C'} \mathbb{E}_{\lambda, \lambda'}\left[(1 - (4\delta)^4)^{\frac{mR}{2}} (1 - (4\delta)^8)^{\frac{mR'}{2}} e^{32 C'^2 \cdot R}\right]$$

$$\leqslant (1 + o(1)) e^{128 \cdot C'} \mathbb{E}_{\lambda, \lambda'}\left[e^{32 C'^2 \cdot R}\right],$$

where the equality follows from the definition of $z_1, z_2$. To conclude, we will use the fact that, for every $k \geqslant 0$,

$$\Pr[R > k] \leqslant \frac{1}{k!}, \tag{23}$$

which was established in Canonne et al. [2020, p.46]. By summation by parts, one can show that this implies

$$\mathbb{E}_R\left[e^{\alpha R}\right] \leqslant 1 + (1 - e^{-\alpha})(e^{e^\alpha} - 1) \xrightarrow[\alpha \to 0]{} 1 + \alpha(e - 1)$$

for any $\alpha > 0$, and so, in our case,

$$\mathbb{E}_{\theta, \theta'}[(1 + \tau(\theta, \theta'))^m] \leqslant (1 + o(1)) e^{128 \cdot C'}\left(1 + (1 - e^{-32 C'^2})(e^{e^{32 C'^2}} - 1)\right) \tag{24}$$

In particular, the RHS can be made arbitrarily close to 1 by choosing a small enough value for the constant $C'$ (in the bound for $m$). By (14), this implies the desired bound on $\|Q - U^{\otimes m}\|_1^2$, and thus establishes Lemma 2.3. $\qquad \square$

**The remaining technical lemma.** It only remains to establish Lemma 2.5, which we do now.

**Lemma 2.5.** *There exists an absolute constant $\delta_0 \approx 0.96$ such that the following holds. Let $K \geqslant 1$ and $R_1, \ldots, R_K$ be integers, and $\delta_1, \ldots, \delta_K \in (0, \delta_0]$. Suppose that $\kappa_{j,1}, \ldots, \kappa_{j,D} \sim \mathrm{Bin}\left(R_j, \frac{1}{2}\right)$, are i.i.d., and mutually independent across $1 \leqslant j \leqslant K$, and $z_j := \frac{1+\delta_j}{1-\delta_j}$. Then*

$$\mathbb{E}\left[\left(\frac{1}{D}\sum_{i=1}^{D}\prod_{j=1}^{K} z_j^{\kappa_{j,i}}\right)^m\right] \leqslant \left(\prod_{j=1}^{K} z_j^{\frac{m}{2}R_j}\right) \mathbb{E}_{\vec{\alpha}}\left[\prod_{i=1}^{D}\prod_{j=1}^{K} \cosh(2\alpha_i \delta_j)\right],$$

*where $(\alpha_1, \ldots, \alpha_D)$ follows a multinomial distribution with parameters $m$ and $(1/D, \ldots, 1/D)$.*

*Proof of Lemma 2.5.* We will require the following simple fact, which follows from the multinomial theorem and the definition of the multinomial distribution:

**Fact A.1.** *Let $D$ be a positive integer and $m$ be a non-negative integer. For any $x_1, \ldots, x_D \in \mathbb{R}$, we have*

$$\left( \frac{1}{D} \sum_{i=1}^{D} x_i \right)^m = \mathbb{E}_{\alpha_1, \ldots, \alpha_D} \left[ \prod_{i=1}^{D} x_i^{\alpha_i} \right],$$

*where $(\alpha_1, \ldots, \alpha_D)$ follows a multinomial distribution with parameters $m$ and $(1/D, \ldots, 1/D)$.*

We now apply Fact A.1 inside the expectation of the LHS of the statement. Note that the sets of random variables $\vec{\alpha} = \{\alpha_1, \ldots, \alpha_D\}$, $\vec{\kappa}_1 = \{\kappa_{1,1}, \ldots, \kappa_{1,D}\}, \ldots, \vec{\kappa}_K = \{\kappa_{K,1}, \ldots, \kappa_{K,D}\}$ are mutually independent, since $\vec{\alpha}$ are a set of auxiliary random variables derived from an averaging operation and by the assumption on $\vec{\kappa}_j$; and we have that $\kappa_{j,1}, \ldots, \kappa_{j,D}$ are i.i.d.,

$$\mathbb{E}\left[ \left( \frac{1}{D} \sum_{i=1}^{D} \prod_{j=1}^{K} z_j^{\kappa_{j,i}} \right)^m \right] = \mathbb{E}\left[ \mathbb{E}_{\alpha_1, \ldots, \alpha_D} \left[ \prod_{i=1}^{D} \left( \prod_{j=1}^{K} z_j^{\kappa_{j,i}} \right)^{\alpha_i} \right] \right]$$

$$= \mathbb{E}_{\vec{\alpha}, \vec{\kappa}_j} \left[ \prod_{i=1}^{D} \prod_{j=1}^{K} z_j^{\alpha_i \kappa_{j,i}} \right]$$

$$= \mathbb{E}_{\vec{\alpha}} \left[ \mathbb{E}_{\vec{\kappa}_j} \left[ \prod_{i=1}^{D} \prod_{j=1}^{K} z_j^{\alpha_i \kappa_{j,i}} \right] \right]$$

$$= \mathbb{E}_{\vec{\alpha}} \left[ \left[ \prod_{i=1}^{D} \prod_{j=1}^{K} \mathbb{E}_{\vec{\kappa}_j} [z_j^{\alpha_i \kappa_{j,i}}] \right] \right]$$

$$= \mathbb{E}_{\vec{\alpha}} \left[ \left[ \prod_{i=1}^{D} \prod_{j=1}^{K} \left( \frac{1 + z_j^{\alpha_i}}{2} \right)^{R_j} \right] \right]$$

(Probability-Generating Function of a Binomial)

$$= \mathbb{E}_{\vec{\alpha}} \left[ \prod_{i=1}^{D} \prod_{j=1}^{K} z_j^{\frac{\alpha_i R_j}{2}} \left( \frac{z_j^{-\alpha_i/2} + z_j^{\alpha_i/2}}{2} \right)^{R_j} \right]$$

$$= \left( \prod_{j=1}^{K} z_j^{\frac{m R_j}{2}} \right) \mathbb{E}_{\vec{\alpha}} \left[ \prod_{i=1}^{D} \prod_{j=1}^{K} \left( \frac{z_j^{-\alpha_i/2} + z_j^{\alpha_i/2}}{2} \right)^{R_j} \right].$$

Next, we will simplify the expression left inside by upper bounding it, using the fact that, given our assumption on $\delta_j$ being bounded above by $\delta_0$, we have $z_j = \frac{1+\delta_j}{1-\delta_j} \leqslant e^{4\delta_j}$. Thus,

$$\mathbb{E}_{\vec{\alpha}} \left[ \prod_{i=1}^{D} \prod_{j=1}^{K} \left( \frac{z_j^{-\alpha_i/2} + z_j^{\alpha_i/2}}{2} \right)^{R_j} \right] \leqslant \mathbb{E}_{\vec{\alpha}} \left[ \prod_{i=1}^{D} \prod_{j=1}^{K} \left( \frac{e^{-2\alpha_i \delta_j} + e^{2\alpha_i \delta_j}}{2} \right)^{R_j} \right]$$

$$= \mathbb{E}_{\vec{\alpha}} \left[ \prod_{i=1}^{D} \prod_{j=1}^{K} (\cosh(2\alpha_i \delta_j))^{R_j} \right]$$

as claimed. $\qquad\qquad\qquad\qquad\qquad\qquad\qquad\qquad\qquad\qquad\qquad\qquad\qquad\qquad\qquad\square$

## A.2 Product Distributions Are Far from Mixture of Trees (Lemma 2.4)

In this subsection, we outline the proof of Lemma 2.4. Our argument starts with Lemma A.2, which allows us to relate the total variation distance between the mixture and the product of its marginals to a simpler quantity, the difference between two components of this mixture.

**Lemma A.2.** *Let $p$ be a distribution on $\{0,1\}^N \times \{0,1\}^M$ (with $N, M \geqslant 2$), and denote its marginals on $\{0,1\}^N$, $\{0,1\}^M$ by $p_1, p_2$ respectively. Then, if $p_1$ is uniform,*

$$d_{\mathrm{TV}}(p, p_1 \otimes p_2) \geqslant d_{\mathrm{TV}}(p(\cdot \mid x_1 = 0), p(\cdot \mid x_1 = 1)).$$

This in turn will be much more manageable, as the parameters of these two mixture components are independent, and thus analyzing this distance can be done by analyzing Binomial-like expressions. This second step is reminiscent of [Canonne et al., 2020, Lemma 8], which can be seen as a simpler version involving only one Binomial instead of two:

**Lemma A.3.** *There exist $C_1, C_2 > 0$ such that the following holds. Let $\varepsilon \in (0,1]$ and $n \geqslant C_1$, and let $a, b$ be two integers such that $a + b = n$ and $b \geqslant \frac{1}{4}n$. Then, for $\delta := \frac{\varepsilon}{\sqrt{n}}$, we have*

$$\frac{(1-\delta)^n}{2^n} \sum_{k_1=0}^{a} \sum_{k_2=0}^{b} \binom{a}{k_1}\binom{b}{k_2} \left| \left( \left(\frac{1+\delta}{1-\delta}\right)^{k_1+k_2} - \left(\frac{1+\delta}{1-\delta}\right)^{k_1+b-k_2} \right) \right| \geqslant C_2 \varepsilon.$$

This parameter $b$ corresponds to the difference between the orientations parameters $\mu, \mu'$ being large, which happens with high constant probability as long as $n$ is large enough. The proof of Lemma A.3 is deferred to Appendix C.3, and we hereafter proceed with the rest of the argument. For fixed $\theta$ and $x_2, \ldots, x_d, z := \frac{1+4\delta}{1-4\delta}$. We will denote by $\mu, \mu'$ the two (randomly chosen) orientation parameters corresponding to the mixture components indexed by $(0, x_2, \ldots, x_d)$ and $(1, x_2, \ldots, x_d)$. By Lemma A.2 and Lemma 1.5, for any product distribution $q$,

$$2d_{\mathrm{TV}}(p, q) \geqslant \frac{1}{3 \cdot 2^N}(1 - 4\delta)^{\frac{N}{2}} \sum_{x_{d+1}, \ldots, x_n} |z^{c(\lambda, \mu, x)} - z^{c(\lambda, \mu', x)}| \tag{25}$$

Let $S_1$ denote the set of pairs in the child nodes with common parameters between $\mu$ and $\mu'$, and $S_2$ the set of pairs with different parameters (that is, the definition of $S_1, S_2$ is essentially that of $A$ and $B$ from the previous section (p. 15), but for equal matching parameters $\lambda = \lambda'$). In particular, we have that $|S_2| = \mathrm{Hamming}(\mu, \mu') \sim \mathrm{Binomial}\left(\frac{N}{2}, \frac{1}{2}\right)$ and $|S_1 \cup S_2| = \frac{N}{2}$. Let $\tilde{c}(S, \mu, x)$ be the analogue of $c_{\mathrm{ch}}(\lambda, \mu, x)$ from (13), but only on a subset of pairs $S$ instead of $\{d, \ldots, n\}^2$; i.e.,

$$\tilde{c}(S, \mu, x) := \left| \left\{ (i, j) \in S : \exists k \in \mathbb{N}, \lambda_k = (i - d + 1, j - d + 1) \text{ and } (-1)^{x_i + x_j} = (-1)^{\mu_k} \right\} \right|.$$

Given any $x, \mu$ and $\mu'$, the following holds from the definitions of $\tilde{c}$ and $c_{\mathrm{ch}}$:

- Since $S_1 \cup S_2$ contains all the pairs, $c_{\mathrm{ch}}(\lambda, \mu, x) = \tilde{c}(S_1, \mu, x) + \tilde{c}(S_2, \mu, x)$ (similarly for $\mu'$).
- Since $S_1$ (resp., $S_2$) contains exactly the pairs whose orientation is the same (resp., differs) between $\mu$ and $\mu'$, we have $\tilde{c}(S_1, \mu', x) = \tilde{c}(S_1, \mu, x)$ and $\tilde{c}(S_2, \mu', x) + \tilde{c}(S_2, \mu, x) = |S_2|$
- For a fixed matching and a partition $S_1, S_2$ of its $N/2$ pairs, given an orientation vector $\mu \in \{0,1\}^{N/2}$, and fixed values $0 \leqslant k_1 \leqslant |S_1|, 0 \leqslant k_2 \leqslant |S_2|$, there are $2^{N/2}\binom{|S_1|}{k_1}\binom{|S_2|}{k_2}$ different vectors $x \in \{0,1\}^N$ such that $\tilde{c}(S_1, \mu, x) = k_1$ and $\tilde{c}(S_2, \mu, x) = k_2$.

Using these properties, we have, assuming $|S_2| \geqslant \frac{1}{4} \cdot \frac{N}{2}$ and $N$ bigger than some constant,

$$\sum_{x \in \{0,1\}^N} |z^{c_{\mathrm{ch}}(\lambda, \mu, x)} - z^{c_{\mathrm{ch}}(\lambda, \mu', x)}| = \sum_{x \in \{0,1\}^N} |z^{\tilde{c}(S_1, \mu, x) + \tilde{c}(S_2, \mu, x)} - z^{\tilde{c}(S_1, \mu', x) + \tilde{c}(S_2, \mu', x)}|$$

$$= \sum_{x \in \{0,1\}^N} |z^{\tilde{c}(S_1, \mu, x) + \tilde{c}(S_2, \mu, x)} - z^{\tilde{c}(S_1, \mu, x) + |S_2| - \tilde{c}(S_2, \mu, x)}|$$

$$= 2^{\frac{N}{2}} \sum_{k_1=0}^{|S_1|} \binom{|S_1|}{k_1} \sum_{k_2=0}^{|S_2|} \binom{|S_2|}{k_2} |z^{k_1 + k_2} - z^{k_1 + |S_2| - k_2}|$$

$$\geqslant C \cdot \frac{2^N}{(1 - 4\delta)^{N/2}} \cdot \varepsilon,$$

where $C > 0$ is an absolute constant, and for the last inequality we invoked Lemma A.3. Recalling now that $|S_2| \sim \mathrm{Bin}\left(\frac{N}{2}, \frac{1}{2}\right)$, for $N$ large enough we also have

$$\Pr\left[|S_2| \geqslant \frac{N}{8}\right] \geqslant 1 - e^{-\frac{N}{16}} > 9/10.$$

Thus, combining the two along with (25), we conclude that

$$\Pr[d_{\mathrm{TV}}(P_\theta(\cdot \mid x_1 = 0, x_2, \ldots, x_d), P_\theta(\cdot \mid x_1 = 1, x_2, \ldots, x_d)) \geqslant \Omega(\varepsilon)] \geqslant 9/10,$$

establishing Lemma 2.4. □

# B Useful results on the MGFs of Binomials and Multinomials

In this section, we establish various self-contained results on the moment-generating functions (MGF) and stochastic dominance of Binomials, truncated (or "capped") Binomials, and multinomial distributions, which we used extensively in Section 2.1 and should be of independent interest. Notably, derivations from Section 2 following (10) are direct consequences of the three lemmas in the section: Lemma B.3, Lemma B.6 and Lemma B.7 below, which we restate and establish later in this section.

**Lemma B.3.** *Let $X \sim \mathrm{Bin}(m, p)$. Then, for any $t$ such that $0 < tm \leqslant 1/16$,*

$$\mathbb{E}[e^{tX^2}] \leqslant \exp(16tm^2p^2 + 2tmp).$$

**Lemma B.6.** *Suppose $\vec{\alpha} = (\alpha_1, \ldots, \alpha_D)^T$ follows a multinomial distribution with parameters $m$ and $(1/D, \ldots, 1/D)$, and $\Delta \geqslant 40D \gg c^4$ be such that $m \leqslant c\sqrt{D}\Delta$. Then, for any $t \leqslant 4$ and $D \geqslant \Omega(1)$, we have*

$$\mathbb{E}\left[\prod_{i=1}^{D} e^{t\alpha_i \mathbb{1}[\alpha_i > \Delta]}\right] \leqslant 1 + c\sqrt{D}\exp\left(-\frac{1}{80}\Delta \log D\right).$$

**Lemma B.7.** *Let $X' \sim \mathrm{Bin}(m, p)$, and $X := \min(X', \Delta)$, for some $\Delta \leqslant m$. Then, for any $t$ such that $0 < t\Delta \leqslant 1/8$ and $0 < tmp \leqslant 1/16$, we have*

$$\mathbb{E}[e^{tX^2}] \leqslant \exp(16tm^2p^2 + 2tmp).$$

## B.1 Bounds on moment-generating functions

We start with some relatively simple statements:

**Fact B.1.** *If $X \sim \mathrm{Bin}(m, p)$, then, for any $0 \leqslant t \leqslant 1$, $\mathbb{E}[e^{tX}] \leqslant \exp(2tmp)$.*

*Proof.* This follows from computing explicitly $\mathbb{E}[e^{tX}] = (1 + p(e^t - 1))^m \leqslant (1 + 2tp)^m \leqslant e^{2tmp}$, where the first inequality uses that $t \leqslant 1$. □

We will also require the following decoupling inequality:

**Lemma B.2.** *Let $F \colon \mathbb{R} \to \mathbb{R}$ be a convex, non-decreasing function, and $X = (X_1, \ldots, X_n)$ be a vector of independent non-negative random variables. Then*

$$\mathbb{E}\left[F\left(\sum_{i \neq j} X_i X_j\right)\right] \leqslant \mathbb{E}\left[F\left(4\sum_{i,j} X_i Y_j\right)\right]$$

*where $Y$ is an independent copy of $X$.*

*Proof.* Introduce a vector of independent (and independent of $X$) $\mathrm{Bern}(1/2)$ random variables $\delta = (\delta_1, \ldots, \delta_n)$; so that $\mathbb{E}[\delta_i(1 - \delta_j)] = \frac{1}{4}\mathbb{1}_{i \neq j}$. For any realization of $X$, we can write

$$\sum_{i \neq j} X_i X_j = 4\mathbb{E}_\delta\left[\sum_{i \neq j} \delta_i(1 - \delta_j)X_i X_j\right] = 4\mathbb{E}_\delta\left[\sum_{i,j} \delta_i(1 - \delta_j)X_i X_j\right],$$

and so, by Jensen's inequality and Fubini, as well as independence of $X$ and $\delta$,

$$\mathbb{E}_X\left[F\left(\sum_{i \neq j} X_i X_j\right)\right] = \mathbb{E}_X\left[F\left(4\mathbb{E}_\delta\left[\sum_{i,j} \delta_i(1 - \delta_j)X_i X_j\right]\right)\right]$$

$$\leqslant \mathbb{E}_\delta\left[\mathbb{E}_X\left[F\left(4\sum_{i,j} \delta_i(1 - \delta_j)X_i X_j\right)\right]\right]$$

This implies that there exists some realization $\delta^* \in \{0,1\}^n$ such that

$$\mathbb{E}_X\left[F\left(\sum_{i \neq j} X_i X_j\right)\right] \leqslant \mathbb{E}_X\left[F\left(4 \sum_{i,j} \delta_i^*(1 - \delta_j^*) X_i X_j\right)\right].$$

Let $I := \{i \in [n] : \delta_i^* = 1\}$. Then $\sum_{i,j} \delta_i^*(1 - \delta_j^*) X_i X_j = \sum_{(i,j) \in I \times I^c} X_i X_j$, and we get

$$\mathbb{E}_X\left[F\left(\sum_{i \neq j} X_i X_j\right)\right] \leqslant \mathbb{E}_X\left[F\left(4 \sum_{(i,j) \in I \times I^c} X_i X_j\right)\right] \tag{26}$$

$$= \mathbb{E}_X\left[F\left(4 \sum_{(i,j) \in I \times I^c} X_i Y_j\right)\right]$$

$$\leqslant \mathbb{E}_X\left[F\left(4 \sum_{(i,j) \in I \times I^c} X_i Y_j + 4 \sum_{(i,j) \notin I \times I^c} X_i Y_j\right)\right]$$

$$= \mathbb{E}_X\left[F\left(4 \sum_{i,j} X_i Y_j\right)\right],$$

where the equality uses the fact that $(X_i)_{i \in I}$ and $(X_j)_{j \in I^c}$ are independent (as $I, I^c$ are disjoint), and so replacing $\sum_{j \in I^c} X_j$ by the identically distributed $\sum_{j \in I^c} Y_j$ does not change the expectation; and the second inequality uses monotonicity of $F$ and non-negativity of $X, Y$, as $4 \sum_{(i,j) \notin I \times I^c} X_i Y_j \geqslant 0$. (Note that up to (and including) (26), the assumption that the $X_i$'s are independent is not necessary; we will use this fact later on.) $\qquad \square$

Note that compared to the usual version of the inequality, we do not require that the $X_i$'s have mean zero; but instead require that they be non-negative, and that $F$ be monotone. We will, in the next lemma, apply Lemma B.2 to the function $F(x) = e^{2tx}$, for some fixed *positive* parameter $t > 0$ (so that $F$ is indeed non-decreasing), and to $X_1, \ldots, X_n$ independent Bernoulli r.v.'s. Specifically, we obtain the following bound on the MGF of the square of a Binomial:

**Lemma B.3.** *Let $X \sim \mathrm{Bin}(m, p)$. Then, for any $t$ such that $0 < tm \leqslant 1/16$,*

$$\mathbb{E}[e^{tX^2}] \leqslant \exp(16tm^2 p^2 + 2tmp).$$

*Proof.* Write $X = \sum_{i=1}^m X_i$, where the $X_i$ are i.i.d. $\mathrm{Bern}(p)$ (in particular, $X_i = X_i^2$). Then, by the Cauchy–Schwarz inequality and the decoupling inequality from Lemma B.2, we have, for $t > 0$,

$$\mathbb{E}[e^{tX^2}] = \mathbb{E}\left[e^{t \sum_i X_i} e^{t \sum_{i \neq j} X_i X_j}\right]$$

$$\leqslant \sqrt{\mathbb{E}\left[e^{2t \sum_i X_i}\right]} \sqrt{\mathbb{E}\left[e^{2t \sum_{i \neq j} X_i X_j}\right]}$$

$$\text{(decoupling)} \quad \leqslant \quad \sqrt{\mathbb{E}\left[e^{2t \sum_i X_i}\right]} \sqrt{\mathbb{E}\left[e^{8t \sum_{i,j} X_i Y_j}\right]}. \tag{27}$$

where $Y_j \sim \mathrm{Bern}(p)$ are i.i.d., and independent of the $X_i$'s. Let $Y = \sum_{i=1}^m Y_i \sim \mathrm{Bin}(m, p)$. From Fact B.1, as long as $2t \leqslant 1$, $8tm \leqslant 1$, and $16tmp \leqslant 1$ (all conditions satisfied in view of our assumption),

$$\mathbb{E}_{X,Y}[e^{8tXY}] = \mathbb{E}_X[\mathbb{E}_Y[e^{8tXY}]] \leqslant \mathbb{E}_X[e^{16tXmp}] \leqslant e^{32tm^2 p^2},$$

and $\mathbb{E}[e^{2tX}] \leqslant e^{4tmp}$. Going back to (27), this implies

$$\mathbb{E}[e^{tX^2}] \leqslant \sqrt{\exp(4tmp)} \sqrt{\exp(32tm^2 p^2)} = \exp(2tmp + 16tm^2 p^2),$$

concluding the proof. $\qquad \square$

We will prove an MGF bound on the truncated Multinomial in Lemma B.6 (noting that using MGF bound of Multinomial distribution is not nearly enough), as required by our analysis on the independence testing lower bound; prior to that, we will need two important lemmas: Lemma B.4 and Lemma B.5. These two lemmas both try to bound the expression with a uniform and more manageable term.

**Lemma B.4.** *Fix $m, \Delta, D$ such that $\frac{m}{\Delta} \leqslant c\sqrt{D}$ for some $c > 0$ (and $D > \max(16c^4, e^{100})$). Fix any integer $k > 0$ and a tuple of non-negative integers $(a_1, \ldots, a_D)$ summing to $m$ such that $L := \sum_{i=1}^{k} a_i > k\Delta$ (in particular, $k \leqslant c\sqrt{D}$). Suppose $(\alpha_1, \ldots, \alpha_D)$ follows a multinomial distribution with parameters $m$ and $(1/D, \ldots, 1/D)$. Then,*

$$e^{4L} \Pr[\vec{\alpha} = (a_1, \ldots, a_D)] \leqslant m \cdot \exp(-\frac{1}{5} L \log D).$$

*Proof.* Via a multinomial distribution grouping argument, the probability can be bounded by considering a grouping of two random variables, $L_1 = \sum_{i=1}^{k} \alpha_i$ and $L_2 = \sum_{i=k+1}^{D} \alpha_i$, where $(L_1, L_2)$ follows a multinomial distribution with parameters $m$ and $(\frac{k}{D}, \frac{D-k}{D})$, namely, recalling $L = \sum_{i=1}^{k} a_i$ and setting $T := \sum_{i=k+1}^{D} a_i$,

$$\Pr[\vec{\alpha} = (a_1, \ldots, a_D)] \leqslant \Pr[L_1 = L, L_2 = T] = \frac{m!}{L!T!} \left(\frac{k}{D}\right)^L \left(\frac{D-k}{D}\right)^T$$

Moreover, note that $m = L + T$. Via Stirling's approximation, we have

$$\frac{m!}{L!T!} \leqslant \exp(m \log m + \log m - L \log L - T \log T) \tag{28}$$

from which we can write, taking the logarithm for convenience,

$$\log\left(\frac{m!}{L!T!}\left(\frac{k}{D}\right)^L\left(\frac{D-k}{D}\right)^T\right) \leqslant \log m - \left(L \log \frac{LD}{mk} + T \log \frac{TD}{m(D-k)}\right) \tag{29}$$

$$= \log m - \left(L \log \frac{LD}{mk} + T \log\left(\frac{TD}{m(D^{3/4}-k)}\frac{D^{3/4}-k}{D-k}\right)\right) \tag{30}$$

$$\leqslant \log m - m \log(D^{1/4}) + (m-L)\log\left(\frac{D-k}{D^{3/4}-k}\right) \tag{31}$$

$$= \log m + m \log\left(1 + k\frac{D^{1/4}-1}{D-kD^{1/4}}\right) - L \log\left(1 + \frac{D^{1/4}-1}{1-k/D^{3/4}}\right)$$

$$\leqslant \log m + mk\frac{D^{1/4}-1}{D-kD^{1/4}} - L \log(D^{1/4}) \tag{32}$$

$$\leqslant \log m + \frac{mk}{D^{1/2}}\frac{1-1/D^{1/4}}{D^{1/4}-c} - \frac{1}{4}L \log D$$

$$\leqslant \log m + \frac{cL}{D^{1/4}-c} - \frac{1}{4}L \log D \tag{33}$$

$$\leqslant \log m + L - \frac{1}{4}L \log D \qquad (\text{as } c/D^{1/4} \leqslant 1/2)$$

where we used Gibbs' inequality for (31); we then have (32) by $\log(1+x) \leqslant x$ for the first term. (33) then follows from $k \leqslant c\sqrt{D}$ and $km \leqslant k\Delta \cdot c\sqrt{D} \leqslant L \cdot c\sqrt{D}$. Finally,

$$e^{4L}\frac{m!}{L!T!}\left(\frac{k}{D}\right)^L\left(\frac{D-k}{D}\right)^T \leqslant \exp(5L - \frac{1}{4}L \log D) \leqslant m \exp\left(-\frac{1}{5}L \log D\right),$$

the last inequality as long as $\log D > 100$. $\qquad \square$

**Lemma B.5.** *Suppose $\vec{\alpha} = (\alpha_1, \ldots, \alpha_D)^T$ follows a multinomial distribution with parameters $m$ and $(1/D, \ldots, 1/D)$, and that $\frac{m}{\Delta} \leqslant c\sqrt{D}$ for some $c > 0, \Delta \geqslant 1$ with $\Delta \geqslant 40D \geqslant \Omega(c^4)$ and*

$D > \Omega(1)$. *For any integer $c\sqrt{D} \geqslant k \geqslant 1$ and any $t \leqslant 4$,*

$$\mathbb{E}\left[\prod_{i:\alpha_i > \Delta} e^{t\alpha_i} \cdot \mathbb{1}[\nu(\vec{\alpha}) = k]\right] \leqslant \exp\left(-\frac{1}{80}\Delta \log(D)\right).$$

*where $\nu(\vec{\alpha}) := |\{i : \alpha_i > \Delta\}|$ denotes the number of coordinates of $\vec{\alpha}$ greater than $\Delta$.*

*Proof.* Without loss of generality, (as later, we will sum over all combinations) assume that $\alpha_1, \ldots, \alpha_k$ are the coordinates larger than $\Delta$, for some integer $k$; and denote their sum by $L$. Note that we then have $k\Delta < L \leqslant m \leqslant c\Delta\sqrt{D}$, and thus $0 \leqslant k \leqslant c\sqrt{D}$.

$$\mathbb{E}\left[\prod_{i:\alpha_i > \Delta} e^{4\alpha_i} \cdot \mathbb{1}[\nu(\vec{\alpha}) = k]\right] = \binom{D}{k} \sum_{\alpha_1,\ldots,\alpha_k > \Delta} \sum_{\alpha_{k+1},\ldots,\alpha_D \leqslant \Delta} e^{4\sum_{i=1}^k \alpha_i} \Pr[\vec{\alpha} = (\alpha_1, \ldots, \alpha_D)] \tag{34}$$

A uniform bound on any $\alpha_1, \ldots, \alpha_D$ as specified can be obtained from Lemma B.4; and, combining it with (34), we have an expression that does not depend on the value of $\vec{\alpha}$; from which[5]

$$
\begin{aligned}
\mathbb{E}\left[e^{4\sum_{i=1}^k \alpha_i}\mathbb{1}[\nu(\vec{\alpha}) = k]\right] &\leqslant \binom{D}{k} \sum_{\alpha_1,\ldots,\alpha_k > \Delta, \alpha_{k+1},\ldots,\alpha_D \leqslant \Delta} m e^{-\frac{1}{5}\Delta \log D} \\
&\leqslant \binom{D}{k}(m - \Delta)^k \Delta^{D-k} \exp\left(\log m - \frac{1}{5}\Delta \log D\right) \\
&\leqslant \exp\left(k \log D + k \log(m - \Delta) + (D - k)\log\Delta + \log m - \frac{\Delta}{5}\log(D)\right) \\
&= \exp\left(k \log(D \cdot \frac{m - \Delta}{\Delta}) + D \log\Delta + \log m - \frac{\Delta}{5}\log(D)\right) \\
&\leqslant \exp\left(-\frac{1}{5}\Delta \log D + (D + 1)\log\Delta + \log(c\sqrt{D}) + \frac{3}{2}k \log(cD)\right) \\
&\leqslant \exp\left(-\frac{1}{10}\Delta \log D + 2c\sqrt{D}\log(cD)\right) \tag{35} \\
&\leqslant \exp\left(-\frac{1}{80}\Delta \log D\right).
\end{aligned}
$$

where (35) follows from $20\frac{D}{\log D} \leqslant \frac{\Delta}{\log \Delta}$, which holds for $\Delta \geqslant 40D$ and $D$ large enough (larger than some absolute constant); and the last inequality holds, given the above constraints, for $D \geqslant 16c^4$. $\quad\square$

**Lemma B.6.** *Suppose $\vec{\alpha} = (\alpha_1, \ldots, \alpha_D)^T$ follows a multinomial distribution with parameters $m$ and $(1/D, \ldots, 1/D)$, and $\Delta \geqslant 40D \gg c^4$ be such that $m \leqslant c\sqrt{D}\Delta$. Then, for any $t \leqslant 4$ and $D \geqslant \Omega(1)$, we have*

$$\mathbb{E}\left[\prod_{i=1}^D e^{t\alpha_i \mathbb{1}[\alpha_i > \Delta]}\right] \leqslant 1 + c\sqrt{D}\exp\left(-\frac{1}{80}\Delta \log D\right).$$

*Proof.* Let $\nu(\vec{\alpha}) := |\{i : \alpha_i > \Delta\}|$ denote the number of coordinates of $\vec{\alpha}$ greater than $\Delta$. Note that $\nu(\vec{\alpha}) < L := \frac{m}{\Delta}$, and that $L = c\sqrt{D}s$ by assumption. We break down the expectation by enumerating over the possible values for $\nu(\vec{\alpha})$, from $0 \leqslant k \leqslant L$:

$$\mathbb{E}\left[\prod_{i=1}^D e^{t\alpha_i \mathbb{1}[\alpha_i > \Delta]}\right] = \mathbb{E}\left[\sum_{k=1}^L \prod_{i:\alpha_i > \Delta} e^{t\alpha_i} \cdot \mathbb{1}[\nu(\vec{\alpha}) = k] + \mathbb{1}[\nu(\vec{\alpha}) = 0]\right]$$

---

[5]We have the number of terms in the summation upper bounded by the following analysis: $(m - \Delta)^k$ is an upper bound of combinations of $\alpha_1, \ldots, \alpha_k$ with values larger than $\Delta$; and similarly, $(\Delta + 1)^{D-k}$ will be the upper bound for the combinations of $\alpha_{k+1}, \ldots, \alpha_D$ with values up to $\Delta$.

$$= \sum_{k=1}^{L} \mathbb{E}\left[\prod_{i:\alpha_i > \Delta} e^{t\alpha_i} \cdot \mathbb{1}[\nu(\vec{\alpha}) = k]\right] + 1 \cdot \Pr[\nu(\vec{\alpha}) = 0]$$

$$\leqslant L \exp\left(-\frac{1}{80}\Delta \log D\right) + \Pr[\nu(\vec{\alpha}) = 0] \tag{36}$$

$$\leqslant cD^{1/2}\exp\left(-\frac{1}{80}\Delta \log D\right) + 1,$$

where (36) follows from Lemma B.5. $\qquad\square$

We now state and prove our last lemma, Lemma B.7, on the MGF of the square of a truncated Binomial:

**Lemma B.7.** *Let $X' \sim \mathrm{Bin}(m, p)$, and $X := \min(X', \Delta)$, for some $\Delta \leqslant m$. Then, for any t such that $0 < t\Delta \leqslant 1/8$ and $0 < tmp \leqslant 1/16$, we have*

$$\mathbb{E}[e^{tX^2}] \leqslant \exp(16tm^2p^2 + 2tmp).$$

*Proof.* We will analyze the sampling process in Definition B.8:

**Definition B.8.** Fix integers $m \geqslant \Delta \geqslant 1$, and let $X'_1, \ldots, X'_m$ be i.i.d. $\mathrm{Bern}(p)$ random variables. Define the distribution of $X_1, \ldots, X_m$ through the following sampling process:

1. Initialize $X_i = 0$ for all $i \in [m]$; sample $\{X'_i\}_{1 \leqslant i \leqslant m}$ as $m$ i.i.d. $\mathrm{Bern}(p)$;
2. If $\sum_{i \in [m]} X'_i < \Delta$, let $X_i = X'_i$ for all $i \in [m]$;
3. If $\sum_{i \in [m]} X'_i \geqslant \Delta$, let $\mathcal{S}' = \{i \in [m] : X'_i = 1\}$ and let $\mathcal{S}$ be a uniformly random subset of $\mathcal{S}'$ of size $\Delta$; set $X_i = X'_i$ for $i \in \mathcal{S}$.

Consider a sequence of random variable $X_1, \ldots, X_m$ as defined in Definition B.8; each $X_i$ (for $1 \leqslant i \leqslant m$) is supported on $\{0, 1\}$ (so that, in particular, $X_i^2 = X_i$); and $X = \sum_{i \in [m]} X_i$. By the Cauchy–Schwarz inequality,

$$\begin{aligned}
\mathbb{E}[e^{tX^2}] &= \mathbb{E}\left[e^{t\sum_{i=1}^{m} X_i + t\sum_{i \neq j} X_i X_j}\right] \\
&\leqslant \sqrt{\mathbb{E}\left[e^{2t\sum_{i=1}^{m} X_i}\right]}\sqrt{\mathbb{E}\left[e^{2t\sum_{i \neq j} X_i X_j}\right]} \\
&\leqslant \sqrt{\mathbb{E}\left[e^{2t\sum_{i=1}^{m} X_i}\right]}\sqrt{\mathbb{E}\left[e^{8t\sum_{(i,j) \in I \times I^c} X_i X_j}\right]} \tag{37} \\
&\leqslant \sqrt{\mathbb{E}\left[e^{2tX}\right]}\sqrt{\mathbb{E}[e^{8tY_1 Y_2}]} \tag{38}
\end{aligned}$$

where $Y_1 \sim \min(\mathrm{Bin}(|I|, p), \Delta)$, $Y_2 \sim \min(\mathrm{Bin}(|I^c|, p), \Delta)$ and $Y_1$ is independent of $Y_2$ (and $(I, I^c)$ is some fixed, but unknown partition of $[m]$). (37) follows from the intermediate step (26) in the proof of Lemma B.2 (observing that $x \mapsto e^{tx}$ is convex, and non-decreasing as $t > 0$; and using the remark from that proof about the independence of $X_i$'s not being required up to that step) and (38) follows from Lemma B.12. We will implicitly use Facts B.9, B.10, and B.11 for the remaining calculations, eventually replacing most expressions with $X' \sim \mathrm{Bin}(m, p)$.

Recalling that $X \leqslant X'$ by definition, the first term of (38) can be bounded as $\mathbb{E}[e^{2tX}] \leqslant e^{4tmp}$. Moreover, from our assumption, $tY_1 \leqslant t\Delta \leqslant 1/8$ and $tmp \leqslant 1/16$. Combined with the fact that $Y_1, Y_2$ is dominated by $X \sim \min(\mathrm{Bin}(m, p), \Delta)$ and thus by $X' \sim \mathrm{Bin}(m, p)$, we have

$$\mathbb{E}[e^{8tY_1 Y_2}] = \mathbb{E}_{Y_1}[\mathbb{E}_{Y_2}[e^{8tY_1 Y_2}]] \leqslant \mathbb{E}_{Y_1}[e^{16tY_1 mp}] \leqslant e^{32tm^2p^2}.$$

Going back to (38), this implies

$$\mathbb{E}[\exp(tX^2)] \leqslant \sqrt{\exp(4tmp)}\sqrt{\exp(32tm^2p^2)} = \exp(2tmp + 16tm^2p^2),$$

concluding the proof. $\qquad\square$

## B.2 Stochastic dominance results between truncated Binomials

**Fact B.9.** *Let* $X \sim \text{Bin}(m, p)$, *and* $0 < n \leqslant m$. *Defining* $Y := \min(X, n)$ *and* $Z := X \mid X \leqslant n$, *we have, for every* $k \geqslant 0$,

$$\Pr[X \geqslant k] \geqslant \Pr[Y \geqslant k] \geqslant \Pr[Z \geqslant k],$$

*i.e.,* $X \succeq Y \succeq Z$, *where* $\succeq$ *denotes first-order stochastic dominance.*

*Proof.* We can write the PMF of $Z$ and $Y$, for all $0 \leqslant k \leqslant n$,

$$\Pr[Y = k] = \begin{cases} \Pr[X = k], & k < n \\ \Pr[X \geqslant n], & k = n \end{cases}, \qquad \Pr[Z = k] = \frac{\Pr[X = k]}{\Pr[X \leqslant n]}.$$

It follows that $\Pr[Y \geqslant k] = \Pr[X \geqslant k]\mathbb{1}\{k \leqslant n\}$, which gives the first part of the statement.

The second part follows from a direct comparison between the two CDF of $Z, Y$: indeed, for $0 \leqslant k \leqslant n$,

$$
\begin{aligned}
\Pr[Y \geqslant k] \geqslant \Pr[Z \geqslant k] \quad &\Leftrightarrow \quad \Pr[X \geqslant k] \geqslant \frac{\Pr[n \geqslant X \geqslant k]}{\Pr[X \leqslant n]} \\
&\Leftrightarrow \quad \Pr[X \geqslant k](1 - \Pr[X > n]) \geqslant \Pr[X \geqslant k] - \Pr[X > n] \\
&\Leftrightarrow \quad \Pr[X \geqslant k]\Pr[X > n] \leqslant \Pr[X > n] \\
&\Leftrightarrow \quad \Pr[X \geqslant k] \leqslant 1,
\end{aligned}
$$

and this last inequality clearly holds. $\qquad\square$

We also record the facts below, which follow respectively from the more general result that first-order stochastic dominance is preserved by non-decreasing mappings, and from a coupling argument.

**Fact B.10.** *Consider two real-valued random variables* $X, Y$, *and* $n \geqslant 0$. *If* $X \succeq Y$, *then* $\min(X, n) \succeq \min(Y, n)$: *for all* $k$,

$$\Pr[\min(X, n) \geqslant k] \geqslant \Pr[\min(Y, n) \geqslant k];$$

*i.e., the* $\min$ *operator preserves first-order stochastic dominance relation.*

**Fact B.11.** *Let* $X \sim \text{Bin}(n, p)$ *and* $Y \sim \text{Bin}(m, p)$, *where* $m \geqslant n$. *Then* $X \preceq Y$.

**Lemma B.12.** *Let* $X_1, \ldots, X_m$ *be sampled from the sampling process in Definition B.8, and* $I, I^c$ *be any partition of* $[m]$. *Define* $Z_I := \sum_{i \in I} X_i$, $Z_{I^c} := \sum_{i \in I^c} X_i$, *and* $Y_I \sim \min(\text{Bin}(|I|, p), n)$, $Y_{I^c} \sim \min(\text{Bin}(|I^c|, p), n)$. *Then*

$$Z_I \cdot Z_{I^c} \preceq Y_I \cdot Y_{I^c}.$$

*Proof.* We prove the lemma by defining a coupling $Z_I, Z_{I^c}, Y_I, Y_{I^c}$ such that $Z_I \cdot Z_{I^c} \leqslant Y_I \cdot Y_{I^c}$ with probability one. The sampling process below will generate samples $(X_i)_{1 \leqslant i \leqslant m}, Z_I, Z_{I^c}, Y_I, Y_{I^c}$ for all possible realizations of $I$ and $I^c$. In other words, from a given sequence $\{X'_i\}_{i \in [m]}$, we will generate $\{X_i\}_{i \in [m]}, Y_{I_1}, Y_{I_1^c}, Y_{I_2}, Y_{I_2^c}, \ldots, Y_{I_{2^m}}, Y_{I_{2^m}^c}, Z_{I_1}, Z_{I_1^c}, Z_{I_2}, Z_{I_2^c}, \ldots, Z_{I_{2^m}}, Z_{I_{2^m}^c}$, where the $(I_i, I_i^c)$ enumerate all partitions of $[m]$ in two sets.

1. Initialize $X_i = 0$ for all $i \in [m]$; sample $(X'_i)_{1 \leqslant i \leqslant m}$ as $m$ i.i.d. $\text{Bern}(p)$;
2. If $\sum_{i \in [m]} X'_i < n$, let $X_i = X'_i$ for all $i \in [m]$;
3. If $\sum_{i \in [m]} X'_i \geqslant n$, let $\mathcal{S}' = \{i \in [m] : X'_i = 1\}$ and let $\mathcal{S}$ be a uniformly random subset of $\mathcal{S}'$ with size $n$; set $X_i = X'_i$ for $i \in \mathcal{S}$.
4. For each $I \in \{I_1, \ldots, I_{2^m}\}$, denote $\mathcal{S}'_I = \mathcal{S}' \cap I$. Select a uniformly random subset of $\mathcal{S}'_I$ with at most $n$ indices which is a superset of $\mathcal{S} \cap I$. In more detail, if $|\mathcal{S} \cap I| < n$, select $\min(|\mathcal{S}'_I|, n) - |\mathcal{S} \cap I|$ elements uniformly at random from $\mathcal{S}'_I \setminus (\mathcal{S} \cap I)$ to add to $\mathcal{S} \cap I$, which becomes $\mathcal{S}_I$; else, let $\mathcal{S}_I = \mathcal{S} \cap I$. Repeat a similar process for $I^c$ to obtain $\mathcal{S}_{I^c}$.
5. For each $I \in \{I_1, \ldots, I_{2^m}\}$, set $Y_I = \sum_{i \in \mathcal{S}_I} X'_i$ and $Y_{I^c} = \sum_{i \in \mathcal{S}_{I^c}} X'_i$.

From the above definition, we can readily see that for any $I$, $Y_I \geqslant Z_I$ and $Y_{I^c} \geqslant Z_{I^c}$. What is left is to argue that the $Y_I \sim \min(\text{Bin}(|I|, p), n)$ and $Y_{I^c} \sim \min(\text{Bin}(|I^c|, p), n)$. We start by noting that for any $k < n$, $\{Y_I = k\} = \{|\mathcal{S}_I| = k\} = \{|\mathcal{S}'_I| = k\}$. The last equality comes from the fact that $|\mathcal{S}_I| < n$ can only mean that $|\mathcal{S}'_I| < n$, and the selection process in step 4 will thus add all elements from $\mathcal{S}'_I$ to $\mathcal{S}_I$. From here, we have $\Pr[Y_I = k] = \Pr[|\mathcal{S}'_I| = k] = \Pr[\text{Bin}(m, p) = k]$, for $k < n$; and we have $\Pr[Y_I = n] = 1 - \Pr[Y_I < n] = \Pr[\text{Bin}(m, p) \geqslant n]$. As a result, $Y_I \sim \min(\text{Bin}(|I|, p), n)$. Similarly, we can argue that $Y_{I^c} \sim \min(\text{Bin}(|I^c|, p), n)$. $\qquad\square$

# C  Deferred Proofs

## C.1  Proofs from Section 3

*Proof of Lemma 3.2.* We analyze Algorithm 2: first we use the algorithm of Lemma 1.3 to learn $P$ to $d_{\chi^2}$ distance $\varepsilon^2$ *as if* it was a product distribution, using $O\left(\frac{n}{\varepsilon^2}\right)$ samples. Let $\hat{P}$ be the output of the learning algorithm. Note that since Lemma 1.3 guarantees *proper* learning, $\hat{P}$ is a product distribution.

We then want to check that $\hat{P}$ is indeed close to $P$ (in Hellinger distance), as it should if $P$ were indeed a product distribution. To do this, we use the algorithm of Theorem 1.4 on $P$, with reference distribution $\hat{P}$ and distance parameter $\varepsilon$; and reject if, and only if, this algorithm rejects.

By a union bound, since both algorithms are correct with probability at least $5/6$, both are simultaneously correct with probability at least $2/3$; we hereafter assume this is the case in our analysis. The total sample complexity is $O\left(n/\varepsilon^2\right) + O\left(\sqrt{2^n}/\varepsilon^2\right) = O\left(2^{n/2}/\varepsilon^2\right)$, as claimed. We now argue correctness.

- *Soundness.* Proof by contrapositive: if the algorithm accepts, this means, from the guarantees of Theorem 1.4, that $d_{\mathrm{H}}(P, \hat{P}) \leqslant \varepsilon$. Since $\hat{P}$ is a product distribution, we conclude that $P$ is $\varepsilon$-close (in Hellinger distance) to being a product distribution.

- *Correctness.* Assume that $P$ is a product distribution. Then, Lemma 1.3 ensures that $d_{\chi^2}(P, \hat{P}) \leqslant \varepsilon^2$; and thus, by Theorem 1.4 the second step will not reject, and the algorithm overall accepts.

Note that, by a standard amplification trick (independent repetition and majority vote), the probability of error can be reduced from $1/3$ to any $\delta \in (0, 1)$ at the price of a $O(\log(1/\delta))$ factor in the number of samples. □

*Proof of Theorem 3.1.* We analyze Algorithm 1, denoting as in the algorithm by $P'$ the product of marginals of $P$, and setting $\delta := \frac{1}{3\binom{n}{d+1}}$, $\varepsilon' := \frac{\varepsilon}{\sqrt{2n}(1+\sqrt{d+1})}$, and

$$m := O\left(\frac{2^{d/2}}{\varepsilon'^2} \log \frac{1}{\delta}\right) = O\left(\frac{2^{d/2}n}{\varepsilon^2} \cdot d^2 \log n\right).$$

The sample complexity is thus immediate; further, note that, as stated in the algorithm, given $m$ i.i.d. samples from $P$ and a fixed set $T \subseteq [n]$ of nodes, one can generate $m$ i.i.d. samples from $P_T$ by only keeping the relevant variables (those in $T$) of each sample from $P$.

- *Completeness.* Assume $P$ is a product distribution. Then, $P_T$ is a product distribution for every choice of $T$, and each of the $\binom{n}{d+1}$ performs thus accepts with probability at least $1-\delta$ by Lemma 3.2. Thus, by a union bound, all tests simultaneously accept with probability at least $1 - \binom{n}{d+1} \cdot \delta = 2/3$, and Algorithm 1 returns "accept."

- *Soundness.* Assume now that $P$ is $\varepsilon$-far (in total variation distance) from every product distribution over $\{0,1\}^n$. A fortiori, it is $\varepsilon$-far from the product distribution $P'$, and thus we have
$$d_{\mathrm{H}}(P, P') \geqslant \frac{1}{\sqrt{2}} d_{\mathrm{TV}}(P, P') \geqslant \frac{\varepsilon}{\sqrt{2}}$$
By Corollary 1.2, this means there exists some node $i \in [n]$ along with the set of its (at most) $d$ parents $\Pi_i$ such that, setting $T := \{i\} \cup \Pi_i$,
$$d_{\mathrm{H}}^2(P_T, P'_T) \geqslant \frac{\varepsilon^2}{2n}.$$
Now, we can invoke our localization lemma, Lemma 3.3, to conclude that $P_T$ is not only far from $P'_T$, it is far from *every* product distribution on $T$:
$$\min_{Q \text{ product}} d_{\mathrm{H}}^2(P_T, Q) \geqslant \frac{\varepsilon^2}{2n(1 + \sqrt{d+1})^2} = \varepsilon'^2.$$

Thus, when this particular set $T$ of $d+1$ nodes is encountered by the algorithm, the corresponding independence test will reject with probability at least $1 - \delta$ by Lemma 3.2, and the overall algorithm will thus reject with probability at least $1 - \delta$.

This concludes the proof: the sample complexity is $O(d^2 2^{d/2} n \log(n)/\varepsilon^2)$ as claimed, and the tester is correct in both cases with probability at least $2/3$. $\qquad \square$

## C.2 Proof of Lemma 3.3

**Lemma 3.3.** *Let $P$ be a distribution on $\{0,1\}^n$, and $P' = (\pi_1 P) \otimes \cdots \otimes (\pi_n P)$ be the product of marginals of $P$. Denoting by $\mathcal{Q}$ the set of all product distributions on $\{0,1\}^n$, we have $\min_{Q \in \mathcal{Q}} d_{\mathrm{H}}(P, Q) \geqslant \frac{1}{1+\sqrt{n}} d_{\mathrm{H}}(P, P')$.*

*Proof.* Since squared Hellinger distance is an $f$-divergence, by the data processing inequality, we have that
$$d_{\mathrm{H}}(P, Q) \geqslant d_{\mathrm{H}}(\pi P, \pi Q). \tag{39}$$
By the subadditivity of Hellinger for Bayes nets from Corollary 1.2 along with (39), we obtain the following:
$$
\begin{aligned}
d_{\mathrm{H}}(P, P') &\leqslant d_{\mathrm{H}}(P, Q) + d_{\mathrm{H}}(Q, P') \\
&\leqslant d_{\mathrm{H}}(P, Q) + \Big( \sum_{i=1}^{n} d_{\mathrm{H}}^2(P'_{X_i}, Q_{X_i}) \Big)^{1/2} \\
&= d_{\mathrm{H}}(P, Q) + \Big( \sum_{i=1}^{n} d_{\mathrm{H}}^2(P_{X_i}, Q_{X_i}) \Big)^{1/2} \\
&\leqslant d_{\mathrm{H}}(P, Q) + \sqrt{n d_{\mathrm{H}}^2(P, Q)} \\
&= \big(1 + \sqrt{n}\big) d_{\mathrm{H}}(P, Q).
\end{aligned}
$$
$\qquad \square$

## C.3 Proof of Lemma A.3

**Lemma A.3.** *There exist $C_1, C_2 > 0$ such that the following holds. Let $\varepsilon \in (0, 1]$ and $n \geqslant C_1$, and let $a, b$ be two integers such that $a + b = n$ and $b \geqslant \frac{1}{4} n$. Then, for $\delta := \frac{\varepsilon}{\sqrt{n}}$, we have*
$$
\frac{(1-\delta)^n}{2^n} \sum_{k_1=0}^{a} \sum_{k_2=0}^{b} \binom{a}{k_1} \binom{b}{k_2} \left| \left( \left( \frac{1+\delta}{1-\delta} \right)^{k_1+k_2} - \left( \frac{1+\delta}{1-\delta} \right)^{k_1+b-k_2} \right) \right| \geqslant C_2 \varepsilon.
$$

*Proof.* By concentration of Binomials,
$$
\left( \frac{1}{2} \right)^n (1-\delta)^n \sum_{k_1=0}^{a} \sum_{k_2=0}^{b} \binom{a}{k_1} \binom{b}{k_2} \left| \left( \left( \frac{1+\delta}{1-\delta} \right)^{k_1+k_2} - \left( \frac{1+\delta}{1-\delta} \right)^{k_1+b-k_2} \right) \right|
$$
$$
\geqslant \left( \frac{1}{2} \right)^n (1-\delta)^n \sum_{k_1=\frac{a}{2}+\sqrt{a}}^{\frac{a}{2}+2\sqrt{a}} \sum_{k_2=\frac{b}{2}+\sqrt{b}}^{\frac{b}{2}+2\sqrt{b}} \binom{a}{k_1} \binom{b}{k_2} \left| \left( \left( \frac{1+\delta}{1-\delta} \right)^{k_1+k_2} - \left( \frac{1+\delta}{1-\delta} \right)^{k_1+b-k_2} \right) \right|
$$
$$
\geqslant \left( \frac{1}{2} \right)^n \frac{C}{\sqrt{ab}} \cdot 2^{|a|+|b|} \sum_{k_1=\frac{a}{2}+\sqrt{a}}^{\frac{a}{2}+2\sqrt{a}} \sum_{k_2=\frac{b}{2}+\sqrt{b}}^{\frac{b}{2}+2\sqrt{b}} (1-\delta)^n \left| \left( \left( \frac{1+\delta}{1-\delta} \right)^{k_1+k_2} - \left( \frac{1+\delta}{1-\delta} \right)^{k_1+b-k_2} \right) \right|
$$
$$
\geqslant \frac{C}{\sqrt{ab}} \sum_{k_1=\frac{a}{2}+\sqrt{a}}^{\frac{a}{2}+2\sqrt{a}} \sum_{k_2=\frac{b}{2}+\sqrt{b}}^{\frac{b}{2}+2\sqrt{b}} (1-\delta)^n \left| \left( \left( \frac{1+\delta}{1-\delta} \right)^{k_1+k_2} - \left( \frac{1+\delta}{1-\delta} \right)^{k_1+k_2+b-2k_2} \right) \right|
$$

where $C$ is some constant larger than 0. For $k_1 + k_2 - \frac{n}{2} = l \in \left[\sqrt{a} + \sqrt{b}, 2\left(\sqrt{a} + \sqrt{b}\right)\right]$, $l \geqslant \sqrt{a+b} = \sqrt{n}$.

$$(1 - \delta)^n \left| \left( \left(\frac{1+\delta}{1-\delta}\right)^{k_1+k_2} - \left(\frac{1+\delta}{1-\delta}\right)^{k_1+k_2+b-2k_2} \right) \right|$$

$$= \left| \left( (1-\delta)^n \left(\frac{1+\delta}{1-\delta}\right)^{n/2+l} - (1-\delta)^n \left(\frac{1+\delta}{1-\delta}\right)^{n/2+l+b-2k_2} \right) \right|$$

$$= (1-\delta^2)^{n/2} \left(\frac{1+\delta}{1-\delta}\right)^l \left| 1 - \left(\frac{1+\delta}{1-\delta}\right)^{b-2k_2} \right|$$

$$\geqslant e^{-\delta^2 n} e^{2\delta \cdot l} \left| 1 - \left(\frac{1+\delta}{1-\delta}\right)^{b-2k_2} \right| \geqslant e^{2\varepsilon - \varepsilon^2} \left| 1 - \left(\frac{1+\delta}{1-\delta}\right)^{b-2k_2} \right| \tag{40}$$

$$\geqslant e^\varepsilon \left( 1 - \left(\frac{1+\delta}{1-\delta}\right)^{b-2k_2} \right) \geqslant e^\varepsilon (1 - e^{4\delta(b-2k_2)}). \tag{41}$$

where (40) and (41) follows from $1 - x \geqslant e^{-2x}$, $0 < x < 0.79$ and $e^{4x} \geqslant \left(\frac{1+x}{1-x}\right) \geqslant e^{2x}$, $0 < x < 0.95$. All these inequalities hold for $n$ larger some constant and every $\varepsilon \in (0, 1]$. Since $b \geqslant \frac{1}{4}n$, and by the summation above $b - 2k_2 \in \left[-4\sqrt{b}, -2\sqrt{b}\right]$,

$$e^\varepsilon (1 - e^{4\delta(b-2k_2)}) \geqslant e^\varepsilon \left( 1 - e^{-8\frac{\varepsilon}{\sqrt{n}}\sqrt{b}} \right) \geqslant e^\varepsilon (1 - e^{-4\varepsilon}) \geqslant \varepsilon.$$

and therefore, summing up every term, we have our lower bound

$$\frac{C}{\sqrt{ab}} \sum_{k_1 = \frac{a}{2} + \sqrt{a}}^{\frac{a}{2} + 2\sqrt{a}} \sum_{k_2 = \frac{b}{2} + \sqrt{b}}^{\frac{b}{2} + 2\sqrt{b}} (1 - \delta)^n \left| \left( \left(\frac{1+\delta}{1-\delta}\right)^{k_1+k_2} - \left(\frac{1+\delta}{1-\delta}\right)^{k_1+k_2+b-2k_2} \right) \right| \geqslant C\varepsilon$$

concluding the proof. $\qquad \square$

### C.4 Proof of (7)

**Fact C.1.** *For any set of cycles such that $\sum_i |\sigma_i| \leqslant n$, we have*

$$\prod_{\substack{\sigma_i : \text{even} \\ |\sigma_i| \geqslant 4}} (1 + (-4\delta)^{|\sigma_i|}) \prod_{\substack{\sigma_i : \text{odd} \\ |\sigma_i| \geqslant 4}} (1 - (-4\delta)^{|\sigma_i|}) \leqslant e^{O\left(\varepsilon^5/n^{\frac{3}{2}}\right)} \prod_{\substack{\sigma_i : \text{even} \\ |\sigma_i| = 4}} (1 + (4\delta)^4) \prod_{\substack{\sigma_i : \text{odd} \\ |\sigma_i| = 4}} (1 - (4\delta)^4)$$

*Proof.* We have

$$\prod_{\sigma_i : \text{even}} (1 + (-4\delta)^{|\sigma_i|}) \prod_{\sigma_i : \text{odd}} (1 - (-4\delta)^{|\sigma_i|})$$

$$= \prod_{\substack{\sigma_i : \text{even} \\ |\sigma_i| \geqslant 5}} (1 + (-4\delta)^{|\sigma_i|}) \prod_{\substack{\sigma_i : \text{odd} \\ |\sigma_i| \geqslant 5}} (1 - (-4\delta)^{|\sigma_i|}) \prod_{\substack{\sigma_i : \text{even} \\ |\sigma_i| = 4}} (1 + (-4\delta)^{|\sigma_i|}) \prod_{\substack{\sigma_i : \text{odd} \\ |\sigma_i| = 4}} (1 - (-4\delta)^{|\sigma_i|})$$

$$\leqslant \prod_{\sigma_i : |\sigma_i| > 5} (1 + (4\delta)^{|\sigma_i|}) \prod_{\sigma_i : \text{even}, |\sigma_i| = 4} (1 + (-4\delta)^{|\sigma_i|}) \prod_{\sigma_i : \text{odd}, |\sigma_i| = 4} (1 - (-4\delta)^{|\sigma_i|})$$

$$\leqslant (1 + (4\delta)^5)^{\frac{n}{4}} \prod_{\sigma_i : \text{even}, |\sigma_i| = 4} (1 + (-4\delta)^{|\sigma_i|}) \prod_{\sigma_i : \text{odd}, |\sigma_i| = 4} (1 - (-4\delta)^{|\sigma_i|})$$

$$\leqslant e^{(4\delta)^5 n/4} \prod_{\sigma_i : \text{even}, |\sigma_i| = 4} (1 + (-4\delta)^{|\sigma_i|}) \prod_{\sigma_i : \text{odd}, |\sigma_i| = 4} (1 - (-4\delta)^{|\sigma_i|})$$

$$\leqslant e^{256\varepsilon^5/n^{\frac{3}{2}}} \prod_{\sigma_i : \text{even}, |\sigma_i| = 4} (1 + (-4\delta)^4) \prod_{\sigma_i : \text{odd}, |\sigma_i| = 4} (1 - (-4\delta)^4)$$

$\qquad \square$

# D Structured Testing Lower Bound

Letting $D = 2^n$, we will rely on the construction from the "standard" lower bound of Paninski [2008] by picking a uniformly random subset $S$ of $\{0,1\}^n$ of size $\frac{D}{2}$. Denote $\mathcal{S}$ the set of all such combinations of $S$, and define $\mathcal{P}_{\mathrm{no}}$ to be $\mathcal{P}_{\mathrm{no}} := \left\{ P = \frac{1+C\varepsilon}{2} U_S + \frac{1-C\varepsilon}{2} U_{S^c} \mid S \in \mathcal{S} \right\}$, where $C > 0$ is a suitable normalizing constant. As before, $U_S$ denotes the uniform distribution on the set of variable $S$ and $P \in \mathcal{P}_{\mathrm{no}}$ is a mixture of two uniform distributions on disjoint parts, with different weights.

It is known that $\Omega(2^{n/2}/\varepsilon^2)$ samples are required to distinguish between such a randomly chosen $P$ and the uniform distribution $U$; further, assume we know that the uniform distribution $U$ is in $\mathcal{C}$. What remains to show is the *distance*, that is, "most" choices of $P \in \mathcal{P}_{\mathrm{no}}$ are $\varepsilon$-far from $\mathcal{C}$. To argue that last part, we will use our assumption that $\mathcal{C}$ can be learned with $m$ samples to conclude by a counting argument: i.e., we will show that there can be at most $2^{mn}$ or so "relevant" elements of $\mathcal{C}$, while there are at least $2^{2^{\Omega(n)}}$ $\mathcal{P}_{\mathrm{no}}$ that are $\varepsilon$-far from each other. Suitably combining the two will establish the theorem below:

**Theorem D.1.** *Let $\mathcal{C}$ be a class of probability distributions over $\{0,1\}^n$ such that the following holds: (1) the uniform distribution belongs to $\mathcal{C}$ (2) there exists a learning algorithm for $\mathcal{C}$ with sample complexity $m = m(n, \varepsilon)$. Then, as long as $mn \ll 2^{O(n)}$, testing whether an arbitrary distribution over $\{0,1\}^n$ belongs to $\mathcal{C}$ or is $\varepsilon$-far from every distribution in $\mathcal{C}$ in total variation distance requires $\Omega(2^{n/2}/\varepsilon^2)$ samples.*

*Proof.* As discussed above, indistinguishability follows from the literature [Paninski, 2008], and thus all we need to show now is that $\mathcal{P}_{\mathrm{no}}$ is far from every distribution in $\mathcal{C}$. By assumption (2), there exists an algorithm $H \colon \{0,1\}^{mn} \to \mathcal{P}$ (without loss of generality, we assume $H$ deterministic) that can output an estimated distribution given $m = m(n, \varepsilon)$ samples from $P \in \mathcal{C}$. Thus, for every $P \in \mathcal{C}$ given $m$ i.i.d. samples $X \in \{0,1\}^{mn}$, $\Pr_{X \sim P^{\otimes m}}(d_{\mathrm{TV}}(H(X), P) < \varepsilon) \geqslant 2/3$.

In particular, this implies the weaker statement that, for every $P \in \mathcal{C}$, there exists *some* $x$ in $\{0,1\}^{mn}$ s.t. $P \in B(H(x), \varepsilon)$ (where $B(x, r)$ denotes the TV distance ball of radius $r$ centered at $x$). By enumerating all possible values in $\{0,1\}^{mn}$, we then can obtain an $\varepsilon$-cover $\{H(x_1), \ldots, H(x_{2^{mn}})\}$ of $\mathcal{C}$, that is, such that $\mathcal{C} \subseteq \bigcup_{i=1}^{2^{mn}} B(H(x_i), \varepsilon)$. The $\varepsilon$-covering number of $\mathcal{C}$ is thus upper bounded by $2^{O(mn)}$.

Next, we lower bound the size of $\mathcal{P}_{\mathrm{no}}$ by constructing an $\varepsilon$-packing $P_\varepsilon$, where $P_\varepsilon = \{P_i \in \mathcal{P}_{\mathrm{no}}, i \in \mathbb{N} : d_{\mathrm{TV}}(P_i, P_j) > \varepsilon, i \neq j\}$. For $P, Q \in \mathcal{P}_{\mathrm{no}}$ corresponding to two sets $S, S'$, each of size $\frac{D}{2} = 2^{n-1}$, we have

$$d_{\mathrm{TV}}(P, Q) = \frac{1}{2}|S \triangle S'| \cdot \left| \frac{1+C\varepsilon}{2} - \frac{1-C\varepsilon}{2} \right| \cdot \frac{2}{D} = C\varepsilon \cdot \frac{|S \triangle S'|}{D} > \varepsilon$$

For this to be at least $\varepsilon$, the pairrwise symmetric difference of (the sets corresponding to the) distributions in $P_\varepsilon$ should be at least $\frac{D}{C} = \Omega(2^n)$. We know, by e.g., Blais et al. [2019, Proposition 3.3] that there exist such families of balanced subsets of $\{0,1\}^n$ of cardinality at least $\Omega(2^{2^{\rho n}})$, where $\rho > 0$ is a constant that only depends on $C$.

Thus, the size of $\mathcal{P}_{\mathrm{no}}$ is itself $\Omega(2^{2^{\rho n}})$; combining this lower bound with the upper bound on the covering number of $\mathcal{C}$ concludes the proof. $\qquad\square$

As a corollary, instantiating the above to the class $\mathcal{C}$ of degree-$d$ Bayes nets over $n$ nodes readily yields the following:

**Corollary D.2.** *For large enough $n$, testing whether an arbitrary distribution over $\{0,1\}^n$ is a degree-$d$ Bayes net or is $\varepsilon$-far from every such Bayes net requires $\Omega(2^{n/2}/\varepsilon^2)$ samples, for any $d = o(n)$ and $\varepsilon \geqslant 2^{-O(n)}$.*

*Proof.* We can obtain a learning upper bound of $m = O(2^d n \log(2^{d+1} n) \log(n^{dn})/\varepsilon^2)$ for degree-$d$ Bayes nets by combining the known-structure case (proven in Bhattacharyya et al. [2020]) with the reduction from known-structure to unknown-structure (via hypothesis selection/tournament [Canonne

et al., 2020]). We have $mn \leqslant O(2^d n^6 / \varepsilon^2)$. To have $2^{mn} \ll 2^{2^{\rho n}}$, where $\rho$ is some constant, we need $mn < 2^{O(n)}$, which requires $d = o(n)$ and $\varepsilon \geqslant 2^{-O(n)}$ for large enough $n$. $\qquad\square$

## D.1 An $\Omega(2^{d/2}\sqrt{n}/\varepsilon^2)$ Lower Bound

In this section, we state and prove a simpler, but quantitatively weaker lower bound than Theorem 2.1 for independence testing, Theorem D.3. This simpler lower bound is adapted from Canonne et al. [2020, Theorem 13] – the "mixture-of-products" construction. Their analysis readily provides indistinguishability, and distance *from the uniform distribution*. Thus, all we need here is to show that most of these hard instances (i.e., "mixtures of products") are far from every product distribution (Lemma D.4), not just the uniform distribution. While the $\Omega(2^{d/2}\sqrt{n}/\varepsilon^2)$ lower bound this yields is not as tight in terms of sample complexity, with a $\sqrt{n}$ dependence instead of $n$ (at a high level, this is because we fix the Bayesian structure, and thus the algorithms have additional information they can leverage), the restriction on $d$ is much milder than the one in Theorem 2.1, allowing up to $d = n/2$.

**Theorem D.3.** *Let $1 \leqslant d \leqslant n/2$. Testing whether an arbitrary degree-$d$ Bayes net over $\{0,1\}^n$ is a product distribution or is $\varepsilon$-far from every product distribution requires $\Omega(2^{d/2}\sqrt{n}/\varepsilon^2)$ samples. This holds even if the structure of the degree-$d$ Bayes net is known.*

*Proof.* As discussed above, we will use the same "mixture-of-products" construction as in Canonne et al. [2020, Theorem 13], which established a lower bound of $\Omega(2^{d/2}\sqrt{n}/\varepsilon^2)$ samples to distinguish it from the uniform distribution. We first recall the definition of this "mixture-of-products" construction.

Letting $N := n - d \geqslant n/2$, we define, for $z \in \{\pm 1\}^N$ the product distribution $p_Z$ over $\{0,1\}^N$ by

$$p_z(x) = \prod_{i=1}^{N}\left(\frac{1}{2} + z_i(-1)^{x_i}\delta\right), \qquad x \in \{0,1\}^N. \tag{42}$$

where $\delta := \frac{\varepsilon}{\sqrt{N}} = \Theta\left(\frac{\varepsilon}{\sqrt{n}}\right)$. A mixture-of-products distribution is then defined by choosing $2^d$ i.i.d. $Z_1, \ldots, Z_{2^d} \in \{\pm 1\}^N$ uniformly at random, and setting $\mathbf{p}_{Z_1,\ldots,Z_{2^d}}$ to be the distribution over $\{0,1\}^n$ which is uniform on the first $d$ bits, and where the first $d$ bits of $x$ are seen as the binary representation (i.e., a "pointer") for which $\mathbf{p}_{Z_i}$ will be used for the last $N$ bits of $x$. That is,

$$\mathbf{p}_{Z_1,\ldots,Z_{2^d}}(x) = \frac{1}{2^d}p_{Z_{\iota(x_1,\ldots,x_d)}}(x_{d+1},\ldots,x_n), \qquad x \in \{0,1\}^n \tag{43}$$

where, analogously to Definition 2.2, $\iota\colon \{0,1\}^d \to [2^d]$ is the indexing function, mapping the binary representation (here on $d$ bits) to the corresponding number.

As mentioned in the preceding discussion, this construction was already used in Canonne et al. [2020, Theorem 13], where the authors show an $\Omega(2^{d/2}\sqrt{n}/\varepsilon^2)$ sample complexity lower bound to distinguish a uniformly randomly chosen mixture-of-products distribution (which is a degree-$d$ Bayes net) from the uniform distribution (which is a product distribution). For their theorem (a lower bound on testing *uniformity*), they then conclude from the easy fact that every such mixture-of-products distribution is $\varepsilon$-far from the uniform distribution. This is not enough for us, as, to obtain the lower bound stated in Theorem D.3, what we need is to show that every such mixture-of-products distribution (or at least *most* of them) is far from *every* product distribution, not just the uniform one. This is the only missing part towards proving Theorem D.3, and is established in our next lemma:

**Lemma D.4** (Distance from Product distributions). *For $p$ uniformly sampled from the mixture-of-products construction,*

$$\Pr\left[\min_{q_1,q_2} d_{\mathrm{TV}}(p, q_1 \otimes q_2) \geqslant \frac{\varepsilon}{750}\right] \geqslant \frac{9}{10}$$

*as long as $n \geqslant d + C_1$, for some constants $C_1 > 0$ and $n/2 \geqslant d$.*

This lemma will directly follow from Claim D.5 (below) and Lemma I.5; the rest of this appendix is thus dedicated to proving the former, which states that most mixture-of-products distributions are far from the product of their marginals.

**Claim D.5.** *Given a mixture-of-products distribution $p$ as in (43), let $p_1$ be the marginal of $\mathbf{p}$ on the first $d$ variables (parent nodes) and $p_2$ the marginal on the $N$ last variables (child nodes). Note that $p_1 \otimes p_2$ is then a product distribution on $\{0,1\}^n$. Then, we have*

$$\Pr\left[d_{\mathrm{TV}}(p, p_1 \otimes p_2) \geqslant \frac{\varepsilon}{250}\right] \geqslant \frac{9}{10}$$

*as long as $n \geqslant C_1 + d$, for some constants $C_1 \geqslant 0$ and $n \geqslant 2d$.*

*Proof.* Fix any mixture-of-products distribution $p$. From Lemma A.2 and the structure of $p$ as given in (43), one can show that

$$d_{\mathrm{TV}}(p, p_1 \otimes p_2) \geqslant \frac{1}{2^{d-1}} \sum_{x_2,\ldots,x_d} d_{\mathrm{TV}}(p(\cdot \mid 0, x_2, \ldots, x_d), p(\cdot \mid 1, x_2, \ldots, x_d)).$$

Denoting $p(\cdot \mid 0, x_2, \ldots, x_d)$ by $p_{\iota(x_2,\ldots,x_d)}$ and $p(\cdot \mid 0, x_2, \ldots, x_d)$ by $q_{\iota(x_2,\ldots,x_d)}$ (where $\iota(x_2, \ldots, x_d) \in [2^{d-1}]$, abusing slightly the definition of the indexing function to extend it to $d-1$ bits), we can rewrite this as

$$d_{\mathrm{TV}}(p, p_1 \otimes p_2) \geqslant \frac{1}{2^{d-1}} \sum_{t=1}^{2^{d-1}} d_{\mathrm{TV}}(p_t, q_t) =: \overline{d_{\mathrm{TV}}}.$$

Now, since $d \leqslant n/2$, one can show that, for every fixed $t$,

$$\Pr[d_{\mathrm{TV}}(p_t, q_t) < \varepsilon/25] < e^{-C \cdot N} \tag{44}$$

where the probability is taken over the choice of $p$ (i.e., its $2^d$ parameters $Z_1, \ldots, Z_{2^d}$), and $C > 0$ is an absolute constant. We defer the proof of this inequality to the end of the appendix, and for now observe that the RHS is less than $1/10$ for $N$ greater than some (related) absolute constant $C_1 > 0$. We can then write, letting $D := 2^{d-1}$ and $X_t := \mathbb{1}\{d_{\mathrm{TV}}(p_t, q_t)\}$

$$\overline{d_{\mathrm{TV}}} \geqslant \frac{\varepsilon}{25} \cdot \frac{1}{2^{d-1}} \sum_{t=1}^{2^{d-1}} \mathbb{1}\left\{d_{\mathrm{TV}}(p_t, q_t) \geqslant \frac{\varepsilon}{25}\right\} = \frac{\varepsilon}{25} \cdot \frac{1}{D} \sum_{t=1}^{D} X_t,$$

where the $X_t$'s are i.i.d. Bernoullis with, by the above analysis, parameter $\alpha \geqslant 1 - e^{-C \cdot N} \geqslant 9/10$. We then have

$$\Pr[d_{\mathrm{TV}}(p, p_1 \otimes p_2) < \varepsilon/250] \leqslant \Pr[\overline{d_{\mathrm{TV}}} < \varepsilon/250] \leqslant \Pr\left[\sum_{t=1}^{D} X_t < \frac{D}{10}\right],$$

so it remains to show that the RHS is less than $1/10$. Since $\mathbb{E}[X_t] \geqslant 9/10$ for all $t$, this readily follows from a Hoeffding bound, for $d \geqslant 1$. $\qquad\square$

To conclude, we only need to prove (44), which (slightly rephrasing it) tells us that two independent parameterizations $p_Z, p'_Z$ will be at total variation distance at least $\Omega(\varepsilon)$ far with overwhelming probability.

*Proof of (44).* Let distribution $p_Z, p_{Z'}$ be defined as in (42), and $Z, Z'$ be i.i.d. and uniform on $\{\pm 1\}^N$. The statement to show is then

$$\Pr\left[d_{\mathrm{TV}}(p_Z, p_{Z'}) \geqslant \frac{\varepsilon}{25}\right] \geqslant 1 - e^{-N/18}, \tag{45}$$

We know (see, e.g., Kamath et al. [2019, Lemma 6.4]) that as long as $\delta \leqslant 1/6$ (which holds for $n \geqslant 36$), then the TV distance is related to the $\ell_2$ distance between mean vectors $\mu_Z, \mu_{Z'} \in [0,1]^N$ as

$$d_{\mathrm{TV}}(p_Z, p_{Z'}) \geqslant \frac{1}{20} \sqrt{\sum_{i=1}^{N} (\mu_{Z,i} - \mu_{Z',i})^2}. \tag{46}$$

Relating this $\ell_2$ distance between mean vectors the Hamming distance between $Z$ and $Z'$, we have

$$\sqrt{\sum_{i=1}^{N}(\mu_{Z,i} - \mu_{Z',i})^2} = \sqrt{\sum_{i=1}^{N}\left(\left(\frac{1}{2} - Z_i\delta\right) - \left(\frac{1}{2} - Z'_i\delta\right)\right)^2} = 2\delta\sqrt{\text{Hamming}(Z, Z')}, \quad (47)$$

where $\text{Hamming}(Z, Z') = \sum_{i=1}^{N}\mathbb{1}[Z_i \neq Z'_i]$. Noting that $\text{Hamming}(Z, Z') \sim \text{Bin}(N, 1/2)$, we have the following via Hoeffding's inequality along with (46) and (47),

$$\Pr\left[d_{\text{TV}}(p_Z, p_{Z'}) \geqslant \frac{1}{20}\sqrt{\frac{4\delta^2 N}{3}}\right] \geqslant \Pr[\text{Hamming}(Z, Z') \geqslant N/3] \geqslant 1 - e^{-N/18}.$$

Since $n/2 \geqslant d$ and thus, $\sqrt{\frac{4\delta^2 N}{3}} \geqslant \frac{4}{5}\varepsilon$, we get (45). $\qquad\square$

This concludes the proof of Lemma D.4, and with it of Theorem D.3. $\qquad\square$