# OpenReview forum: "Independence Testing for Bounded Degree Bayesian Networks"
_NeurIPS.cc/2022/Conference — NeurIPS 2022 Accept_

### Official Review · Reviewer_wLb1 · 2022-07-10

**Rating:** 5
**Confidence:** 3
**Soundness:** 3 good
**Presentation:** 2 fair
**Contribution:** 3 good

**Summary:**

This study focuses on the independence test where a given bayeisan network distribution is a product distribution or it is epsilon-far in total variation distiance from any product distribution. This study shows that the required sampale size is O(2^{d/2} n/ epsilon^2) rather than exp(n) using the sparsity of a directed acyclic graph.


**Questions:**

1. Unclear main problem: This paper states that the main problem is whether the joint distribution of a Bayesian network is a production distribution. However, as stated on the page 4 in the paper, it satisfies the factorization property, and hence, the joint distribution is a product distribution. It would be better to clarify the focused problem. I am not sure if I understand correctly, is the proposed tester can distinguish P from a bayesian network with X_1 -> X_2 -> X_3, and Q from a bayesian network with X_1 <- X_2 <- X_3?
2. This paper does not assume the known structure. However, I do not understand how this study can apply the unknown ordering of a graph, which is in general not solvable.
3. This paper misuses the term degree as in-degree. This makes the paper is really unclear. The maximum indegree, d_{in} = \max_{j \in V} | \Pi_j |,  the maximum degree, d = \max_{j \in V} | \Pi_j \cup {child of j} |,, and the maximum degree of the moralized graph is d_{m} = \max_{j \in V} | \Pi_j \cup {child of j} \cup {spouse of j} |.
4. It would be better to discuss that how to a given a distribution is from a sparge Bayesian netowrk?
5. For the enhanced clarity, the theoretical findings would be better to be confrimed through numerical experiments.

**Ethics Review Area:**

["I don’t know"]

**Limitations:**

None.

**Strengths And Weaknesses:**

Strenghts: Better sample complexity.
Weakeness: Some unclear statements.

---

> ### Author Response · Authors · 2022-08-02
> **Response for wLb1**
>
> We thank the reviewer for their feedback, and respond in detail to the points they raised below. We hope that they address the reviewer’s concerns, and will lead to their reevaluating their score.
>
> 1. We would like to clarify that while both a Bayes net and a product distribution satisfy certain factorization properties, they are not equivalent, e.g., a distribution P is a Bayes net with bounded in-degree $d$ does not necessarily imply that it is a product distribution (the converse is true). Specifically, the factorization property the reviewer refers to (p.4 in the paper) expresses the joint distribution as a product of *conditional* distributions; while for a product distribution, this is just the product of the individual marginals. Put differently, the question we address is whether a Bayes net with maximum in-degree $d$ (unknown graph structure) can be expressed as a Bayes net with maximum in-degree 0.
> \
> Regarding the second question: no, our tester would not distinguish between the two cases (and, more generally, we are not addressing the question of structure learning or identification). Our tester would distinguish between either of the two cases (degree-1 Bayes nets) and the corresponding product distribution on $X_1, X_2, X_3$ (no edges), assuming the edges in the proposed example encode non-trivial dependencies (i.e., removing them significantly changes the resulting distribution).
>
> 2. While we do not know the structure (graph) of the Bayes net, for fixed $n,d$, the number of possible graphs is finite, and therefore, it is possible to iterate through them; the resulting algorithm is computationally inefficient, but as stated in the paper our focus here is the *sample complexity*.
>
> 3. We defined the notion of degree used (bound on the maximum in-degree) on l.172. We followed here the literature on learning and testing Bayesian networks (see, e.g., https://arxiv.org/abs/2011.04144 or https://proceedings.mlr.press/v65/canonne17a.html); we will clarify this by adding “in-degree” explicitly when defining the degree, on lines 172-173.
>
> 4. We are not sure to understand what the reviewer is suggesting. If the suggestion is the (related, but different) question of estimating how far to a sparse Bayes net a given distribution is, we note that this is orthogonal to our current paper (we work under the assumption of sparse Bayes net, not trying to test that assumption), but also that this is partially addressed in Theorem D.1, where we show that even testing if a distribution is *exactly* a sparse Bayes net requires exponentially many samples (in the dimension $n$).
>
> 5. The main focus of this paper is on the theoretical analysis and understanding of the sample complexity of the problem, and especially the lower bound. We believe our algorithms to be practical, and agree that an extensive evaluation of this aspect would be interesting. However, we leave this for future work, as this would distract from the main message of the current paper.
>
> We are happy to engage in more discussion if the reviewer has further questions.

---

### Official Review · Reviewer_Fnfu · 2022-07-10

**Rating:** 6
**Confidence:** 3
**Soundness:** 4 excellent
**Presentation:** 2 fair
**Contribution:** 3 good

**Summary:**

This paper shows that for a distribution $P$ over $n$ binary variables such that it factorizes according to an underlying degree-$d$ Bayesian network, testing that $P$ is a product law versus $P$ is $\varepsilon$-away from any product law in total variation, requires $O(2^{d/2} n / \varepsilon^2 )$ samples. Further, the bound is shown to be tight.

**Questions:**

1. The problem studied should be better motivated.

(i) Why focused on testing complete independence (i.e., product measure)? Perhaps elaborate on *testing graphical structure* --- the problem is testing that the underlying Bayesian network has no edge, I suppose?

(ii) Why focusing on the total variation distance? Does the result also extend to the problem posed in Hellinger distance?

(iii) Does this problem bear any practical implication?

2. Relation to Daskalakis et al. (2019) for Ising models.

The relation is dismissed as "Ising models and Bayes nets are incomparable modeling assumptions". But certain Ising models are Bayesian networks, e.g., when the undirected graph is chordal. Perhaps, at least for these cases, it is worth more clarification?

3. Total variation vs. Hellinger

This issue arises again when reading the high-level description of techniques in $\S$1.2, where the challenge seems to be that while previous testing algorithms are developed for total variation, the localization result in Daskalakis and Pan (2017) is in Hellinger. The authors chose to stick with Hellinger and devised a testing algorithm for Hellinger.

My question is about the alternative: can the localization result be extended to the total variation?

4. Definition 2.2 is confusing.

I am confused by the following.

(1) $\lambda$ is disconnected (as a perfect matching) but called a "tree".

(2) The expression for $\text{Cov}(X_i, X_j)$ does not make sense to me: $(-1)^{\mu_{l,k}} = (-1)^{\pm 1}$ is always $-1$?

(3) What does "pointer" mean here?

In general, it is very difficult for me to parse how the construction works. For better clarity, I would suggest moving the text description from page 3 to follow the definition. Or, even better, illustrate with an example.

5. The proof sketch in $\S$2.1 is too dense.

It should be reduced and simplified to make it a readable **sketch**. Meanwhile, the paper should end properly with some discussion, which is currently impossible due to space limit.

**Limitations:**

I do not foresee any potential negative societal impact of their work.

I have listed my suggestions in the previous section.

**Strengths And Weaknesses:**

### Strengths
1. The paper solves the problem studied: the bound is tight.
2. The problem and techniques are connected to and built upon many recent developments, e.g., Diakonikolas and Kane (2016), Daskalakis and Pan (2017), and  Canonne et al. (2020).
3. The proofs seem solid.

### Weakness
1. The authors should better motivate the problem studied, e.g., why focusing on testing complete independence, why using the total variation, etc.
2. The presentation of certain sections can be improved.

---

> ### Author Response · Authors · 2022-08-02
> **Response for reviewer Fnfu**
>
> We thank the reviewer for their time and useful comments, especially for pointing out some typos in the paper (their question 4 (2)) and for their feedback and suggestions on the presentation of our lower bound constructions. Please find our response, clarifications and corrections below:
>
> 1. (i) This is a great question — indeed, testing independence is a special case of testing graphical structure (testing the Bayes net has in-degree 0 on every node, or as the reviewer writes, testing if the network has no edge). We refer the reviewer to our motivation section in line 55-60: by answering this (fundamental) question of independence testing, we are making an important step towards the general (and even more challenging) problems of testing whether a sparse network has even sparser connections (not just the sparsest, as in the case of independence testing). Put differently, a long-term goal is to address the more general question, as stated in ll. 55-60; yet, even the “base case” of independence testing was completely open prior to our work, and is a first and necessary step towards a full understanding of the general case. We hope that our paper will inspire others to study related graphical testing problems, e.g., testing the maximum in-degree of $k$, where $k \ll d$.
> \
>  (ii) Total variation distance is a standard metric used in hypothesis testing, due to its relation to Type I and II error (Pearson–Neyman lemma), indistinguishability, and its other properties (data processing inequality, relation to Hellinger/KL/$\chi^2$/$L_2$, etc.) … It does make sense, in some cases, to consider other notions of distances (see, e.g., reference [Daskalakis, Kamath, and Wright. *Which distribution distances are sublinearly testable?*]), often as a proxy for TV distance or in combination to obtain some additional robustness. We note that our upper bound result, while stated for TV, does imply a result for Hellinger distance as well, since our analysis relies on Hellinger distance as a proxy for TV.
> \
> (iii) While our paper has a strong theoretical flavor, we strongly believe that our algorithm and result can have some implications for a range of hypothesis testing questions, given the fundamental nature of the problem solved and the pervasiveness of the sparse Bayes net assumption/modelling in natural sciences and science at large. We refer the reviewer to lines 37-43 of the submission for some pointers to such applications.
>
> 2. Thank you for pointing this out; indeed, in some special cases, the results overlap, though the conversion between parameterizations makes the results difficult to compare even in these cases (e.g., dependence on the parameter $\beta$ (maximum edge value) in the results for the Ising model case, and max-undirected-degree vs. max-in-degree). We will add a more thorough discussion and comparison in the final version.
>
> 3. There are analogous localization results on TV distance. But these translate to much weaker results compared to the ones provided in Daskalakis and Pan (2017) -- using the localization results for TV, we would get worse dependency in $\varepsilon$, which due to our setting of that parameter when using that result also then leads to a worse dependence on $n$. From a technical standpoint, the issue with trying to use or derive such localization results for TV distance directly is that TV distance does not “tensorize” (while, say, Hellinger, $\chi^2$, and KL do): *even for product distributions*, TV does not tensorize, and there is no tight localization result possible: so, obtaining a sufficiently tight one (for our purposes) for more general Bayes net structures is impossible.
>
> 4. (1) Thank you for the feedback: we will clarify this point in the final version. Namely, we use the term “tree” to refer to the structure of degree-$1$ Bayes nets (as in the lower bound construction), while this is indeed a forest (the max degree is 1). This technically still factorizes as a Bayesian path (tree), but not all edges are necessary.
> \
>  (2) It is a typo on our end; indeed, it should be $\\{0, 1\\}$.
> \
> (3) By “pointer”, we mean that the first $d$ nodes with a support of $\\{0,1\\}^d$, can be seen as a $d$-bit string that acts as a pointer to one of the $2^d$ distributions, i.e., the value of the first $d$ nodes encodes which distribution the rest of the $(n-d)$ nodes are on. We will clarify this in the final version, and follow the reviewer’s suggestion about the construction by adding an illustration (figure).
>
>  5. This is a fair point; we will do our best to make this proof sketch more concise and less dense, and if enough space remains, we will add a short discussion section.
>
> We are happy to engage in more discussion if the reviewer has further questions.

---

> > ### Comment · Reviewer_Fnfu · 2022-08-08
> > **Thanks for the reply**
> >
> > I would like to thank the authors for their detailed reply.
> >
> > I feel my comments have been properly addressed. I would suggest incorporating your reply to certain points into the manuscript, especially on point 1(ii) (why focusing on the total variation) and point 3 (using localization in TV leads to sub-optimal results).

---

> > > ### Author Response · Authors · 2022-08-09
> > > **Thanks for the suggestion...**
> > >
> > > ... we will make sure to incorporate these to the paper.

---

### Official Review · Reviewer_j9Gv · 2022-07-25

**Rating:** 6
**Confidence:** 4
**Soundness:** 3 good
**Presentation:** 3 good
**Contribution:** 3 good

**Summary:**

The paper studies the independence testing problem. Given samples from a distribution $P$ over $n$ binary random variables, one should detect where all $n$ random variables are statistically independent or not (i.e., at least $\epsilon$-far in TV from any product distribution).

For a general $P$'s, the sample complexity is in the order of $\exp(n)$. The main contribution in this paper is to show that when $P$ is a Bayesian network with in-degree at most $d \ll n$, the sample complexity is in the order of $n \exp(d) / \epsilon^2$.

**Questions:**

I would appreciate if the authors would address the following:

1) Please comment on why the problem would be interesting from the viewpoint of applied Machine Learning.  Specifically, why a Machine Learning practitioner would like to run the independence testing algorithm?

2) Learning the structure of Bayesian networks is useful for understanding independence between variables (e.g., genes, brain regions, etc.) in exploratory research. Why are Bayesian networks useful in the independence testing problem (besides the fact that it leads to a better sample complexity)?

3) Besides the bounds on the MGF of squared binomials and the result for the Hellinger distance. Could the authors point out to other specific novel technical results (lemmas, equations, etc.)? Any argument related to the challenge of solving this problem as well as the technical novelty will be highly appreciated.

4) Please clarify if the problem/algorithm assumes that data is observational, or whether the algorithm needs to perform interventions (fixing some variables and sample other variables). See for instance "Use the $m$ samples from $S$ to generate $m$ i.i.d. samples from $P_T$" in Algorithm 1.

**Limitations:**

As the paper is theoretical, there is no clear and direct societal impact.

**Strengths And Weaknesses:**

Strengths:
- In section 1.2, I truly appreciate that the authors spent a good amount of space comparing their results with relatively simpler alternatives, but that have worse sample complexity than their result.
- I also appreciate how the authors provide some intuition about the novelty of the techniques used in this paper, specifically in Lines 114-120 and Lines 147-156.
- The theoretical result seem sound and thorough.

Weaknesses:
- The theoretical result is interesting, but the (possible) Machine Learning application is unclear.
- It is unclear (at least to me) whether the problem/algorithm assumes observational data or it requires interventions.

---

> ### Author Response · Authors · 2022-08-02
> **Response for reviewer j9Gv**
>
> We are thankful to the reviewer for their reviews and thoughtful comments. Please find our response and clarifications below:
>
> 1. Besides applications to hypothesis testing (such as the ones discussed in the introduction, e.g., checking whether variables are independent in medical settings; see line 37-41), we would give one example scenario for applying our algorithms in the real world, namely, *model selection*: suppose the ML practitioner might have some rough ideas (valid assumptions) on the generation process and would like to leverage this piece of information to save on data collection. Namely, they know, based on well-established prior knowledge of the domain that the data collected does have sparse dependency structures (satisfying our low in-degree assumption) and would like to obtain good estimates of the distribution (density estimation). They also suspect (based on less certain, yet plausible knowledge/modeling assumptions) that the variables are actually statistically independent, i.e., the Bayes net they want to learn might in fact be a product distribution.\
> In this case, they have two strategies: one is to be safe and learn the distribution as a Bayesian Network, which requires (up to constants) $2^d n/\varepsilon^2$ observations. The other is to first check whether that less-certain hypothesis does holds, and, if so, estimate the density with much fewer observations, only $n/\varepsilon^2$ (as we then know that it is close to a product distribution). With our algorithm, the second approach is possible using an additional $2^{d/2} n/\varepsilon^2$ observations, which dominates the total number and leads to significant savings over the first approach.
>
> 2. The fact that Bayesian networks arise naturally in a variety of settings (see ll .37-41) makes this assumption a very natural one. In that sense, these savings (in terms of sample complexity, i.e., data size; and the resulting savings in time and storage to process, collect, and store this data) are paramount. Since this structure is readily available in a lot of applications, having algorithms to leverage it in such a fundamental application as testing statistical independence can be incredibly useful in any data-starved setting. Put differently: if your goal is to perform independence testing, then leveraging all the structural assumptions you know you have is very natural, especially (as we show) since it brings huge savings in data requirements.
>
> 3. We will refer the reviewer to Line 644-648 and onwards, where we use existing and establish new stochastic dominance results (which we believe are of independent interest) to prove the upper bound on the MGF of truncated Binomials. \
> At a higher level, our analysis combines a variety of techniques which, while not separately new per se, are used in a non-trivial way: decoupling inequality, stochastic dominance, and coupling of random variables, for instance. \
> Finally, Theorem D.1 is also a new and very general result on testing, which we believe could find other applications to easily obtain testing lower bounds for other classes of high-dimensional probability distributions from known learning results.
>
> 4. No; no intervention is needed for our algorithm. For “Use the $m$ samples from $S$ to generate $m$ i.i.d. samples from $P_T$" in Algorithm 1”, we are looking at marginals of the full distribution $P$, and are interested in a subset of variables $T$. To get (generate) one sample for $P_T$, we can simply take one sample of $P$, keep the corresponding node values of $T$ and ignore the rest.
>
>
>
> We are happy to engage in more discussion if the reviewer has further questions.

---

### Meta-Review · Area_Chair_wbu5 · 2022-08-28

**Recommendation:** Accept
**Confidence:** Certain

**Metareview:**

The manuscript studies the independence testing problem, given samples from a distribution over several binary random variables. While the sample complexity is exponential (in the number of variables), this paper shows that when the distribution is a Bayesian network with small in-degree, the sample complexity is linear.

All reviewers asked for a clarification in the motivation, and some reviewers asked for comparison to literature and/or possible alternatives. The authors addressed this well during the rebuttal phase. I recommend adding this to the camera-ready version of the paper, as well as other discussions and clarifications raised by all the reviewers.

**Award:**

No

---

### Decision · Program_Chairs · 2022-09-14

Accept